# Elastic and Balanced End-to-end Training of Dynamic LLMs with DynMo

## Abstract

To reduce the computational and memory costs of Large Language Models (LLMs), schemes that introduce dynamic training are increasingly emerging. Examples of dynamic models are: a) Mixture of Experts (MoEs) at which token routing affects the compute balance, b) gradual pruning of the parameters of a model, c) dynamically freezing layers, d) dynamic sparse attention schemes, e) early exit of tokens as they pass through the model layers, and f) Mixture of Depths (MoDs) schemes where tokens bypass blocks. One side effect that limits the practical value of dynamic models is the introduction of workload imbalance among workers, which in turn negatively affects the efficiency in distributed training. We propose a dynamic load balancing solution (DynMo), with a proof that it satisfies maximum reduction in imbalance, to adaptively maintain equal compute workloads among different workers in pipeline parallelism. In addition, DynMo dynamically packs work into fewer workers, while sustaining training throughput, to release the idle workers back to the job manager. DynMo supports both single nodes with multi-GPUs and systems with multi-GPU multi-nodes. In comparison to static distributed training solutions (Megatron-LM and DeepSpeed), DynMo accelerates the end-to-end training of dynamic GPT models by up to 1.23x (MoEs), 3.18x (parameter pruning), 2.23x (layer freezing), 4.02x (sparse attention), 4.52x (early exit), and 1.17x (MoDs). DynMo is available at
https://anonymous.4open.science/r/DynMo-4D04/.

## 1 Introduction

Sizes of neural networks used to train LLMs has exponentially grown since the introdution of transformers Vaswani et al. (2017). This growth demands more memory and compute power. Yet, neither the memory capacity nor the compute capability of a single accelerator increases at the same rate Sevilla et al. (2022). As a result, high-performance computing centers and cloud providers use a mix of model and data parallelism for training large models Narayanan et al. (2021). One of the most commonly used forms of model parallelism in LLMs is pipeline parallelism, in which consecutive layers are grouped into stages, with each stage assigned to one accelerator (worker) Kahira et al. (2021). Input mini-batches are split into micro batches (chunks) to improve accelerator utilization by overlapping computation in a pipeline fashion Huang et al. (2019); Harlap et al. (2018); Fan et al. (2021); Li & Hoefler (2021); Qi et al. (2024).

In traditional LLMs training schemes, the workload for each pipeline stage is known in advance and remains static throughout the training. To reduce computational resource requirements, new training schemes that introduce dynamic training workloads are emerging. This includes: **a)** neural networks where different input samples take different pathways through the model layers [1], e.g. gated neural networks Shazeer et al. (2017a), sparsely activated Mixture of Experts (MoEs) Zhou et al. (2022b), Switch Transformers Fedus et al. (2022), Mixture of Depths (MoDs) Raposo et al. (2024) etc., **b)** gradual pruning where the parameters of a model are pruned (i.e. sparsified) during training Gale et al. (2019), **c)** freeze training where some of the layers of the model are adapatively frozen during training Wang et al. (2022), **d)** different schemes to dynamically sparsify the attention matrix Liu et al. (2022a); Pagliardini et al. (2023); Tay et al. (2020), and **e)** early exit strategies where tokens skip remaining layers based on an exit decision Elbayad et al. (2020); Schuster et al. (2022); Liu et al.

---

[1]In this paper we use *layer* to refer to a transformer block (multi-head attention + a feed forward network)

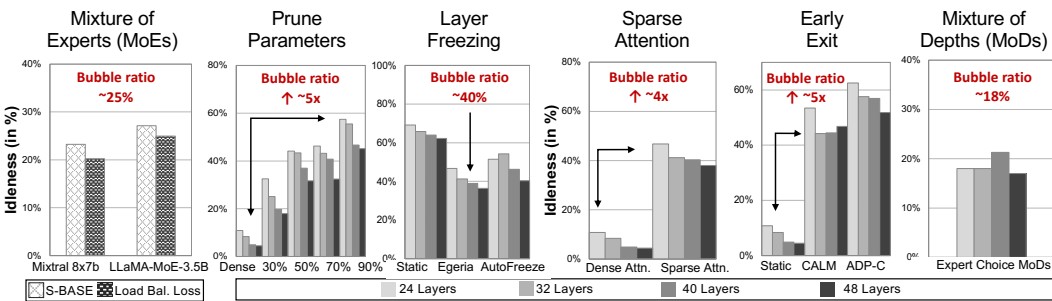

Figure 1: Average idleness percentage of GPUs (per iteration) for training dynamic GPT models Radford et al. (2018). Five example cases use models parameterized to have between 24 and 48 layers, and for one example case (MoEs) we report the average idleness percentage for Mixtral 8x7b and LlaMA-MoE-3.5B models. For pipeline parallelism, we use the highest performing pipeline parallelism scheme known to the authors: the "almost zero-bubble pipeline parallelism" scheme Qi et al. (2024). All reported bubble ratios are measured on a hybrid of pipeline parallelism and data parallelism on 720 H100 GPUs in total, excluding MoEs which uses 128 H100 GPUs in total. **Mixture of Experts**: we observe ∼25% bubble ratio in the pipeline on Mixtral 8x7b Jiang et al. (2024) and LLaMA-MoE-3.5b Team (2024), arising from the load imbalance imposed by the routing schemes used in token choice (S-BASE Lewis et al. (2021b) and load imbalance with auxiliary loss Jiang et al. (2024)). **Gradual prunning of model parameters**: we observe almost a five fold increase in idleness at 90% sparsity levels. Note that idleness at *Dense* is the inherent pipeline bubbles of a static model. **Layer freezing**: SoTA freezing schemes that incorporate load balancing (Egeria Wang et al. (2022) and AutoFreeze Liu et al. (2021)) yield ∼40% bubble ratio. **Dynamic Sparse Flash Attention**: locality sensitive hashing with support for flash attention Pagliardini et al. (2023) exhibits a 4x increase in the bubble ratio over the baseline dense attention. **Early exit**: SoTA early exit methods (CALM Schuster et al. (2022) and ADP-C Liu et al. (2022b)) exhibits up to 5x increase in the bubble ratio over the baseline (w/o early exit), mainly due to the accumulation of bubbles in late layers. **Mixture of Depths**: we observe ∼18% bubble ratio in the pipeline, arising from the load imbalance imposed by the routing scheme of expert choices that lacks information about future tokens Raposo et al. (2024).

(2022b); Kim et al. (2022). Other than computational efficiency, there is a wide range of reasons that motivate the use of different forms of dynamic models to improve certain model attributes, such as explainability and generalization. We refer the reader to the surveys Han et al. (2021); Tay et al. (2022) on different forms of dynamic models.

One of the main downsides of using dynamic models is that they introduce load imbalance in pipeline parallelism, effectively decreasing the throughput of LLM training Zhou et al. (2022a); He et al. (2022). For example, Figure 1 shows the average idleness of GPUs for GPT language models with different numbers of layers, for different types of dynamic models. Load imbalance manifests itself as *bubbles* that appear in the pipeline due to a stalling accelerator waiting to receive work from its late neighboring worker(s). Since a pipeline is only as fast as its slowest stage, load balancing becomes crucial for efficient resource utilization.

Production distributed training solutions typically implement a static load balance at the beginning of training and maintain the same load distribution throughout the training. For instance, Megatron-LM Shoeybi et al. (2019) evenly distributes all transformer layers across the accelerators. DeepSpeed Smith (2023) currently offers three partitioning methods for distributing model layers: *Uniform*, which balances the number of layers; *param*, which balances the number of parameters in each stage; and *regex*, which balances layers whose names match a given regex pattern. However, this approach operates on the assumption that the accelerators' workloads remain roughly unchanged throughout training. As a result, it fails to address the pipeline stalls introduced by dynamic models, ultimately leading to a decrease in computational efficiency.

Considering the increasing importance of efficient MoEs/MoDs, sparse dynamic models, layer freezing, and other dynamic training workloads, this work aims to mitigate the pipeline stalls introduced by dynamic models. We introduce DYNMO, an elastic load-balancing framework designed for

dynamic models, to ensure balanced pipeline stages during training. DYNMO dynamically re-distributes the workload among accelerators whenever an imbalance arises during training, consequently enhancing computational efficiency and leading to cost savings. DYNMO incorporates two different dynamic balancers, both proven to converge to the optimal workload balance among workers. Our experiments demonstrate that DYNMO incurs negligible overhead and can scale effectively in both: a) single-node multi-GPU environments and b) multi-node multi-GPU environments typically used for training LLMs with hybrid parallelism.

DYNMO not only enhances performance through dynamic load balancing but also offers the capability to elastically adapt GPU resources. Specifically, as the total workload decreases during training due to gradual pruning or early exit, the load balancer consolidates the work onto fewer GPUs –subject to memory capacity constraints– while maintaining performance. GPUs that are no longer needed for training can then potentially be released back to the job scheduler.

DYNMO is the first work to study pipeline stalls caused by training dynamic models. Innovative designs of dynamic models struggle to deliver practical impact, at large scale training, unless there is a platform from which those models can be made efficient. DYNMO provides the essential platform for achieving efficiency in these models. Additionally, considering the substantial costs required for each training run of GPT-class models Li (2022); Heim (2022); Morgan (2022), improving the efficiency of dynamic models can result in significant cost savings.

Finally, we emphasize that **DYNMO has no impact on model accuracy, as its role is solely to redistribute workload, without interfering with the learning process or changing the learning regime in any way**, i.e., DYNMO functions as a complementary system software solution that operates independently of the underlying strategies used for parameters pruning, early exist strategy, layer freezing, experts routing etc. This makes DYNMO extendable and compatible with various dynamic schemes. In principle, it can even be applied to models that undergo dynamic changes for reasons other than the six example cases we list in this paper, such as manufacturing variability of computing units Sinha et al. (2022). In short, our contributions are:

- We introduce DYNMO, which enables researchers to explore dynamic models and significantly improves the end-to-end training efficiency of such models, making their practical application more feasible. We invoke DYNMO to rebalance at regular intervals without prior knowledge of dynamism, hence balancing the load in a fully automated and transparent manner. DYNMO is orthogonal to the underlying pipeline parallelism and dynamism scheme; it allows for compatibility with various dynamic compute and model reduction schemes.

- We propose two load balancing algorithms proven to converge to optimal balancing in order to alleviate the negative effects of dynamic models on pipeline utilization. We further introduce a scheme for reducing the number of GPUs used during training by re-packing work to fewer GPUs.

- We show the benefits of the framework with six different example cases of dynamic models, in both single-node and multi-node settings. DYNMO achieves 1.23x (MoEs), 3.18x (parameter pruning), 2.23x (layer freezing), 4.02x (sparse attention), 4.52x (early exit), and 1.17x (MoDs) speedups over (static) Megatron-LM on multi-node hybrid data and pipeline parallelism with up to 720 H100 GPUs. We demonstrate that the re-packing strategy can be effective in reducing the number of GPUs by up to half while sustaining comparable performance.

## 2 MOTIVATION AND BACKGROUND

### 2.1 BUBBLES IN PIPELINE PARALLELISM

There are two types of bubbles in pipeline parallelism: (i) inherent bubbles of the pipeline schedule (e.g. bubbles in-between forward and backward passes in GPipe Huang et al. (2019)), and (ii) bubbles introduced by the dynamic models during training (e.g. bubbles introduced by sparsification during training). We aim to reduce the latter type of bubbles by carefully redistributing the layers among stages to minimize the workload imbalance in the pipeline. Appendix B elaborates with analysis of bubbles in pipeline parallelism.

## 2.2 DYNAMIC MODELS

To reduce the compute and memory costs, training schemes that introduce dynamic training workloads are increasingly emerging. The irregular control-flow in those dynamics models lead inevitably to load imbalance. This leads to inefficiencies that often cause the dynamic model to be slower than the baseline, hence defeating the purpose of using a dynamic model.

The load balancing problem considered can be formally defined as follows. Given a set of workers $\mathcal{N} = \{\mathcal{N}_1, \mathcal{N}_2, \ldots, \mathcal{N}_n\}$ and a set of tasks $\mathcal{T} = \{t_1, t_2, \ldots, t_m\}$, each task $t_j \in \mathcal{T}$ is associated with a workload $c_j$. The total workload is denoted by:

$$C = \sum_{j=1}^{m} c_j.$$

Let $A : \mathcal{T} \to \mathcal{N}$ be a load assignment function that maps each task to a worker. The load of a worker $\mathcal{N}_i \in \mathcal{N}$, denoted $L_i$, is defined as the sum of the workloads of tasks assigned to it:

$$L_i = \sum_{t_j \in A^{-1}(\mathcal{N}_i)} c_j.$$

The objective of the load balancing problem is to minimize the maximum load among all workers:

$$\min_A \max_{i \in \{1, \ldots, n\}} L_i = \min_A \max_{i \in \{1, \ldots, n\}} \left( \sum_{t_j \in A^{-1}(\mathcal{N}_i)} c_j \right).$$

This optimization problem aims to distribute the tasks such that the workload is balanced across the workers, minimizing the worst-case scenario in terms of load.

In the following we list the dynamic model example cases we examine in this paper.

**Mixture of Experts**

In Mixture of Experts (MoEs) Shazeer et al. (2017b), the model capacity is increased by routing input tokens selectively to specialized sub-networks known as experts, rather than processing all tokens through the same feed-forward network. This design improves efficiency but introduces a new source of load imbalance.

We define the load imbalance in MoEs as follows. Let $\mathcal{E} = \{e_1, e_2, \ldots, e_k\}$ be the set of experts in an MoE layer, where each expert $e_i$ is assigned to a worker $\mathcal{N}_i \in \mathcal{N}$. During distributed training, tokens are routed to these experts based on a routing function $R : \mathcal{T} \to \mathcal{E}$, where each token $t_j \in \mathcal{T}$ is sent to one (or more) experts.

The load of an expert $e_i$, denoted $L_{e_i}$, is defined as the total workload of the tokens assigned to it:

$$L_{e_i} = \sum_{t_j \in R^{-1}(e_i)} c_j.$$

Ideally, the load should be balanced across all experts, i.e., $L_{e_i} \approx L_{e_j}$ for all $e_i, e_j \in \mathcal{E}$. However, in practice, the routing function $R$ does not guarantee perfect balance due to the stochastic and learned nature of the routing process, which often includes an auxiliary loss Jiang et al. (2024) or a linear assignment problem Lewis et al. (2021b).

Let $L_{\max}$ and $L_{\min}$ denote the maximum and minimum loads across the experts:

$$L_{\max} = \max_{i \in \{1, \ldots, k\}} L_{e_i}, \quad L_{\min} = \min_{i \in \{1, \ldots, k\}} L_{e_i}.$$

The load imbalance $\Delta L$ is defined as the relative difference between the maximum and minimum loads:

$$\Delta L = \frac{L_{\max} - L_{\min}}{\frac{1}{k} \sum_{i=1}^{k} L_{e_i}}.$$

In practical scenarios, even for large batch sizes, this imbalance can be significant. For example, empirical results from the Mixtral 8x7b model Jiang et al. (2024) demonstrate up to 40% difference between the busiest and least busy experts. This is consistent with the imbalance observed in token choice routing reported by several others (e.g., Zhou et al. (2022b); Raposo et al. (2024). This discrepancy arises because the routing function $R$ often includes a learned component, such as an auxiliary loss attached to the MLPs or a linear assignment scheme, leading to suboptimal and dynamic routing choices.

This imbalance propagates through the training pipeline, causing bubbles in the communication phase, particularly during the all_to_all collective operation, where tokens are returned to their original workers. The result is inefficiency in GPU utilization and increased latency, highlighting the importance of developing improved load balancing mechanisms for MoEs in distributed training systems.

**Parameter Pruning**

Parameter pruning removes a subset of model parameters, resulting in a sparse network that can maintain similar performance to the original dense model. However, pruning during training introduces load imbalance due to non-uniform pruning across different layers.

Parameter pruning methods, such as global magnitude pruning Hagiwara (1993), selectively remove parameters across the entire network based on their importance. Let $\mathcal{L} = \{l_1, l_2, \ldots, l_d\}$ be the set of layers in the neural network, and each layer $l_i$ is assigned to a worker $\mathcal{N}_i \in \mathcal{N}$. During distributed training, the workload of each layer may change dynamically as parameters are pruned.

Let $p_i^{(k)}$ denote the fraction of parameters retained in layer $l_i$ at time step $k$. The effective workload of layer $l_i$ at time step $k$, denoted $c_i^{(k)}$, is proportional to the number of remaining parameters:

$$c_i^{(k)} = p_i^{(k)} \cdot c_i,$$

where $c_i$ is the initial workload of layer $l_i$ before pruning. The total workload of the network at time step $k$ is:

$$C^{(k)} = \sum_{i=1}^{d} c_i^{(k)}.$$

Let $A^{(k)} : \mathcal{L} \to \mathcal{N}$ be the load assignment function at time step $k$. The load of a worker $\mathcal{N}_j \in \mathcal{N}$, denoted $L_j^{(k)}$, is defined as the sum of the effective workloads of the layers assigned to it:

$$L_j^{(k)} = \sum_{l_i \in (A^{(k)})^{-1}(\mathcal{N}_j)} c_i^{(k)}.$$

Ideally, the load should be balanced across all workers, i.e., $L_j^{(k)} \approx L_{j'}^{(k)}$ for all $\mathcal{N}_j, \mathcal{N}_{j'} \in \mathcal{N}$. However, if the pruning method does not prune each layer uniformly (e.g., global magnitude pruning), the fraction $p_i^{(k)}$ may vary significantly across layers, leading to load imbalance.

Let $L_{\max}^{(k)}$ and $L_{\min}^{(k)}$ denote the maximum and minimum loads among the workers at time step $k$:

$$L_{\max}^{(k)} = \max_{j \in \{1, \ldots, n\}} L_j^{(k)}, \quad L_{\min}^{(k)} = \min_{j \in \{1, \ldots, n\}} L_j^{(k)}.$$

The load imbalance $\Delta L^{(k)}$ at time step $k$ is defined as the relative difference between the maximum and minimum loads:

$$\Delta L^{(k)} = \frac{L_{\max}^{(k)} - L_{\min}^{(k)}}{\frac{1}{n} \sum_{j=1}^{n} L_j^{(k)}}.$$

In practice, non-uniform pruning introduces significant differences in layer workloads, causing imbalances that lead to pipeline stalls or bubbles Zhu & Gupta (2017); Frankle & Carbin (2018); Bellec et al. (2017). These bubbles occur because certain workers become idle while waiting for the completion of tasks assigned to workers handling more heavily pruned layers. This dynamic load imbalance emphasizes the need for adaptive load balancing strategies to efficiently handle pruned models in distributed training scenarios. Appendix B elaborates on the factors that drive pruning: pruning criteria, structure, schedule.

**Layer freezing**

Layer freezing is a technique used to reduce computational costs during training by halting updates to certain layers of an LLM once they have converged. While layer freezing can lead to significant efficiency gains, it also introduces load imbalance when the frozen layers are unevenly distributed among workers.

Let $\mathcal{L} = \{l_1, l_2, \ldots, l_d\}$ denote the set of layers in the DNN, where each layer $l_i$ is assigned to a worker $\mathcal{N}_i \in \mathcal{N}$. During training, earlier layers of the network often converge faster Wang et al. (2022), leading to the possibility of freezing these layers to reduce computation. Let $f_i^{(k)} \in \{0, 1\}$ be an indicator variable for layer $l_i$ at time step $k$, where:

$$f_i^{(k)} = \begin{cases} 1, & \text{if layer } l_i \text{ is frozen at time step } k, \\ 0, & \text{otherwise.} \end{cases}$$

The effective workload of a layer $l_i$ at time step $k$, denoted $c_i^{(k)}$, is adjusted based on its frozen status:

$$c_i^{(k)} = (1 - f_i^{(k)}) \cdot c_i,$$

where $c_i$ is the initial workload of layer $l_i$ before freezing. If $f_i^{(k)} = 1$, then $c_i^{(k)} = 0$, indicating that the layer contributes no computational load.

The total workload at time step $k$ is:

$$C^{(k)} = \sum_{i=1}^{d} c_i^{(k)}.$$

Let $A^{(k)} : \mathcal{L} \to \mathcal{N}$ be the load assignment function at time step $k$. The load of a worker $w_j \in \mathcal{W}$, denoted $L_j^{(k)}$, is defined as:

$$L_j^{(k)} = \sum_{l_i \in (A^{(k)})^{-1}(w_j)} c_i^{(k)}.$$

Ideally, the load should be balanced across all workers, i.e., $L_j^{(k)} \approx L_{j'}^{(k)}$ for all $\mathcal{N}_j, \mathcal{N}_{j'} \in \mathcal{N}$. However, if frozen layers are not evenly distributed among workers, this can lead to significant load imbalance.

Let $L_{\max}^{(k)}$ and $L_{\min}^{(k)}$ denote the maximum and minimum loads among the workers at time step $k$:

$$L_{\max}^{(k)} = \max_{j \in \{1,\ldots,n\}} L_j^{(k)}, \quad L_{\min}^{(k)} = \min_{j \in \{1,\ldots,n\}} L_j^{(k)}.$$

The load imbalance $\Delta L^{(k)}$ at time step $k$ is defined as:

$$\Delta L^{(k)} = \frac{L_{\max}^{(k)} - L_{\min}^{(k)}}{\frac{1}{n} \sum_{j=1}^{n} L_j^{(k)}}.$$

In practice, this imbalance occurs because the frozen layers often reside in the earlier parts of the network, which may be unevenly assigned across workers. This uneven distribution can cause certain workers to experience reduced computational load while others handle non-frozen, heavier workloads, resulting in pipeline stalls Shen et al. (2020). As a result, dynamic load balancing strategies are needed to address the changing workload during training when using layer freezing.

**Dynamic Sparse Flash Attention**

Dynamic sparse flash attention is a recent technique that combines hash-based sparse attention with FlashAttention Dao et al. (2024); Pagliardini et al. (2023). This approach leverages dynamic sparsification to accelerate attention computations by restricting the attention matrix to specific blocks determined by hash codes. However, the varying levels of sparsification across layers introduce significant load imbalances during distributed training.

Let $\mathcal{L} = \{l_1, l_2, \ldots, l_d\}$ be the set of layers in the neural network, where each layer $l_i$ employs a dynamic sparse attention mechanism. The workload of each layer depends on the sparsity level induced by the hash-based attention mechanism, which varies dynamically during training.

Define $s_i^{(k)}$ as the sparsity factor of layer $l_i$ at time step $k$, where $s_i^{(k)} \in [0, 1]$ represents the fraction of non-zero elements in the attention matrix after sparsification. The effective workload of layer $l_i$ at time step $k$, denoted $c_i^{(k)}$, is adjusted based on the sparsity factor:

$$c_i^{(k)} = s_i^{(k)} \cdot c_i,$$

where $c_i$ is the initial workload of layer $l_i$ before applying dynamic sparsification. The total workload at time step $k$ is:

$$C^{(k)} = \sum_{i=1}^{d} c_i^{(k)}.$$

Let $A^{(k)} : \mathcal{L} \to \mathcal{N}$ be the load assignment function at time step $k$. The load of a worker $\mathcal{N}_j \in \mathcal{N}$, denoted $L_j^{(k)}$, is defined as:

$$L_j^{(k)} = \sum_{l_i \in (A^{(k)})^{-1}(\mathcal{N}_j)} c_i^{(k)}.$$

In an ideal scenario, the loads should be balanced across all workers, i.e., $L_j^{(k)} \approx L_{j'}^{(k)}$ for all $\mathcal{N}_j, \mathcal{N}_{j'} \in \mathcal{N}$. However, due to the dynamic nature of hash-based sparse attention, the sparsity factor $s_i^{(k)}$ can vary significantly across layers, causing discrepancies in the workloads assigned to different workers.

Let $L_{\max}^{(k)}$ and $L_{\min}^{(k)}$ denote the maximum and minimum loads among the workers at time step $k$:

$$L_{\max}^{(k)} = \max_{j \in \{1, \ldots, n\}} L_j^{(k)}, \quad L_{\min}^{(k)} = \min_{j \in \{1, \ldots, n\}} L_j^{(k)}.$$

The load imbalance $\Delta L^{(k)}$ at time step $k$ is defined as:

$$\Delta L^{(k)} = \frac{L_{\max}^{(k)} - L_{\min}^{(k)}}{\frac{1}{n} \sum_{j=1}^{n} L_j^{(k)}}.$$

In practice, the dynamic sparsification introduced by hash-based attention leads to varying levels of computational effort among different layers and across different time steps Pagliardini et al. (2023). This variability results in significant load imbalances, as some workers may process layers with higher sparsity (lower workload), while others handle layers with denser attention matrices. Consequently, dynamic load balancing mechanisms are required to mitigate these imbalances and maintain efficient utilization of distributed resources during training.

**Early Exit**

Early Exit is a technique that allows tokens to skip subsequent layers once they have reached a confident state. While this method, also known as token pruning, can significantly reduce computational costs, it introduces a new form of load imbalance due to the uneven distribution of token processing across layers.

Let $\mathcal{L} = \{l_1, l_2, \ldots, l_d\}$ denote the set of layers in the neural network, where tokens can exit the model early, skipping the remaining layers. Define $t^{(k)}$ as the total number of tokens being processed at time step $k$, and let $t_i^{(k)}$ be the number of tokens processed by layer $l_i$ at time step $k$. In an Early Exit scheme, $t_i^{(k)}$ generally decreases with the layer depth, as tokens exit before reaching deeper layers Schuster et al. (2022); Liu et al. (2022b).

The effective workload of layer $l_i$ at time step $k$, denoted $c_i^{(k)}$, is proportional to the number of tokens processed by that layer:

$$c_i^{(k)} = \frac{t_i^{(k)}}{t^{(k)}} \cdot c_i,$$

where $c_i$ is the initial workload of layer $l_i$ before applying Early Exit. The total workload at time step $k$ is:

$$C^{(k)} = \sum_{i=1}^{d} c_i^{(k)}.$$

Let $A^{(k)} : \mathcal{L} \to \mathcal{N}$ be the load assignment function at time step $k$. The load of a worker $\mathcal{N}_j \in \mathcal{N}$, denoted $L_j^{(k)}$, is defined as:

$$L_j^{(k)} = \sum_{l_i \in (A^{(k)})^{-1}(\mathcal{N}_j)} c_i^{(k)}.$$

Ideally, the loads should be balanced across all workers, i.e., $L_j^{(k)} \approx L_{j'}^{(k)}$ for all $\mathcal{N}_j, \mathcal{N}_{j'} \in \mathcal{N}$. However, due to Early Exit, the number of tokens processed by deeper layers ($t_i^{(k)}$ for larger $i$) decreases significantly, leading to a reduced workload for workers handling these layers.

Let $L_{\max}^{(k)}$ and $L_{\min}^{(k)}$ denote the maximum and minimum loads among the workers at time step $k$:

$$L_{\max}^{(k)} = \max_{j \in \{1, \ldots, n\}} L_j^{(k)}, \quad L_{\min}^{(k)} = \min_{j \in \{1, \ldots, n\}} L_j^{(k)}.$$

The load imbalance $\Delta L^{(k)}$ at time step $k$ is defined as:

$$\Delta L^{(k)} = \frac{L_{\max}^{(k)} - L_{\min}^{(k)}}{\frac{1}{n} \sum_{j=1}^{n} L_j^{(k)}}.$$

In practice, Early Exit schemes cause severe load imbalances as fewer tokens reach the deeper layers, leading to underutilization of the workers responsible for these layers. The imbalance can be exacerbated, resulting in increased bubble ratios (up to 5x) due to idle time Schuster et al. (2022).

This makes Early Exit an ideal case for dynamic load balancing strategies, such as repacking layers into fewer workers to sustain training throughput.

**Mixture of Depths**

Mixture of Depths(MoDs) Raposo et al. (2024) is a technique that generalizes the early exit scheme by allowing tokens to skip not only the final layers but also intermediate ones. The version of MoDs used in this paper is based on expert choice, leveraging Mixture of Experts (MoEs) for enhanced performance. However, the dynamic nature of layer skipping and the inherent variability in expert choice introduce significant load imbalances during distributed training.

Let $\mathcal{L} = \{l_1, l_2, \ldots, l_d\}$ be the set of layers in the neural network, and $t^{(k)}$ be the total number of tokens being processed at time step $k$. In the Mixture of Depths scheme, tokens can skip intermediate layers based on expert predictions, leading to a varying number of tokens processed at each layer. Let $t_i^{(k)}$ denote the number of tokens processed by layer $l_i$ at time step $k$.

Define $r_i^{(k)}$ as the routing weight for layer $l_i$, determined by an auxiliary MLP predictor that estimates whether a token should bypass that layer or not. The effective workload of layer $l_i$ at time step $k$, denoted $c_i^{(k)}$, is then:

$$c_i^{(k)} = r_i^{(k)} \cdot t_i^{(k)} \cdot c_i,$$

where $c_i$ is the initial workload of layer $l_i$ before applying the MoD scheme. The total workload at time step $k$ is:

$$C^{(k)} = \sum_{i=1}^{d} c_i^{(k)}.$$

Let $A^{(k)} : \mathcal{L} \to \mathcal{N}$ be the load assignment function at time step $k$. The load of a worker $\mathcal{N}_j \in \mathcal{W}$, denoted $L_j^{(k)}$, is defined as:

$$L_j^{(k)} = \sum_{l_i \in (A^{(k)})^{-1}(\mathcal{N}_j)} c_i^{(k)}.$$

In an ideal scenario, the loads would be balanced across all workers, i.e., $L_j^{(k)} \approx L_{j'}^{(k)}$ for all $w_j, w_{j'} \in \mathcal{W}$. However, two main factors contribute to load imbalance in the MoD scheme:

1. **Prediction Inaccuracies**: The auxiliary MLP predictor used to route tokens can misestimate whether a token will be among the top-k selected for the next layer. These inaccuracies lead to fluctuations in $r_i^{(k)}$, causing variability in the token distribution across layers. 2. **Integration with MoEs**: The MoD implementation leverages Mixture of Experts (MoEs) for improved performance, which introduces additional variability in token routing. Since the MoE layers dynamically select experts based on token features, the effective workload $c_i^{(k)}$ may differ significantly from the expected workload, amplifying the imbalance.

Let $L_{\max}^{(k)}$ and $L_{\min}^{(k)}$ denote the maximum and minimum loads among the workers at time step $k$:

$$L_{\max}^{(k)} = \max_{j \in \{1, \ldots, n\}} L_j^{(k)}, \quad L_{\min}^{(k)} = \min_{j \in \{1, \ldots, n\}} L_j^{(k)}.$$

The load imbalance $\Delta L^{(k)}$ at time step $k$ is defined as:

$$\Delta L^{(k)} = \frac{L_{\max}^{(k)} - L_{\min}^{(k)}}{\frac{1}{n} \sum_{j=1}^{n} L_j^{(k)}}.$$

Empirically, we observe imbalances of up to 18% in the MoD scheme due to the combined effects of routing prediction errors and the inherent variability in expert selection in MoEs Raposo et al.

---

**Algorithm 1** End-to-end Training of Dynamic LLMs with DYNMO

---

    **Input:** model, train_iters, rank, workers
    **Input:** dynamism_args, balance_args, pack_args
1: dynamism_rat, dynamism_region, dynamism_freq ← dynamism_args
2: is_load_balance, balancer ← balance_args
3: is_pack, num_workers_to_pack ← pack_args
4: dynamism_idx ← 0
5: dynamism_iter ← NULL
6: profile ← 0
7: **for** iter ← 0 **to** train_iters **do**
8:     train_step(model, profile)
9:     **if** iter **in** dynamism_region AND iter % prune_freq == 0 **then**
10:         worker_ dynamism(model, dynamism_rat[dynamism_idx], rank)      ▷ Algo. 2 in Appendix C
11:         dynamism_idx += 1
12:         dynamism_iter = iter
13:         **if** is_load_balance **then**
14:             profile = 1
15:         **end if**
16:     **end if**
17:     **if** is_load_balance AND iter == dynamism_iter + 1 **then**
18:         load_balance(model, balancer)      ▷ Algo. 3 in Appendix C
19:         profile ← 0
20:     **end if**
21:     **if** is_pack AND iter == dynamism_iter + 1 **then**
22:         pack_workload(model, num_workers_to_pack)      ▷ Algo. 4 in Appendix C
23:     **end if**
24: **end for**

---

(2024). These imbalances result in uneven workloads across workers, necessitating adaptive load balancing strategies to maintain efficient training throughput.

## 3 DYNMO: ELASTIC AND BALANCED END-TO-END TRAINING OF DYNAMIC LLMS

### 3.1 OVERVIEW

In this work, we use six example cases of dynamic models for which current (static) distributed training systems are not ready to handle efficiently. Even though we show the efficiency of our load balancing system for dynamic LLMs with these example cases, they can be a basis for expanding to other forms of dynamic models.

Algorithm 1 shows the overall flow of operations of DYNMO with model dynamism. The dynamism function (line 10) depends on the target case. For instance, if the target case is parameter pruning, the dynamism function would apply global pruning on different worker. The algorithm takes as input a model, the number of training iterations, the rank of the accelerator, and several arguments for managing the dynamism, balancing, and packing the model's workloads. We start the training with the original model and train it until a user-specified dynamism to apply is reached. The frequency varies by the target case, in layer freezing it is as frequent as every 50 iterations, while in parameter tuning the pruning frequency is in the range of 3000-7000 iterations. The model or control flow is altered only if the training is in the dynamism stage. In this stage, the model is adjusted every $dynamism\_freq$ iteration, where the model (or control flow inside the model) is modified until it meets the specified stopping criteria (lines 9-16). The first iteration after each dynamism operation is used for profiling the time it takes to execute each layer in the altered model and the memory usage of all workers (accelerators) in the pipeline. Next, DYNMO collects the profiling information and decides on balancing the workload by moving layers across pipeline stages based on the execution times of individual layers to minimize the pipeline stalls, subject to the constraints of memory capacity per worker (line 17-20). DYNMO also attempts to re-pack the total workload into fewer number of GPUs if the re-packing feature is enabled by the end user (line 21-23). Once the training is out of the dynamism region, the balanced pipeline continues to execute with the model. Figure

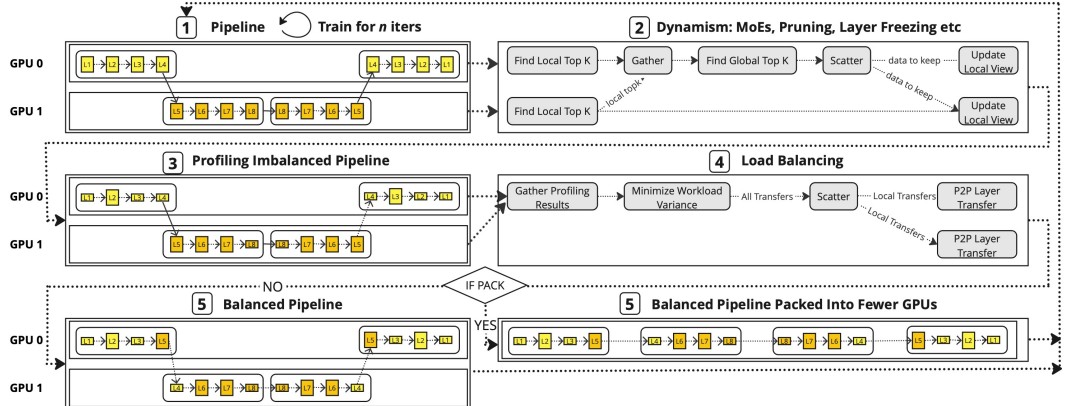

Figure 2: Overview of DYNMO. The flow in the figure (top to bottom) is repeated until training is completed. Each yellow and orange rectangle represents a transformer layer. The size of a rectangle illustrates the amount of work in a layer. (1) shows the pipeline before the model starts to change due to dynamism. (2) some action (dynamically) changes the model, or the flow of work inside the model. (3) profiles the pipeline to check if there is any imbalance between stages, (4) performs load balancing based on the profiling results, (5) trains the balanced pipeline until the next time to rebalance, optionally it reduces the number of resources (GPUs) used in training by re-packing.

2 illustrates the overview of DYNMO with all its steps. The implementation of individual steps of model (or control-flow) altering, load balancing, and re-packing can be found in their respective sections.

## 3.2 PROFILING THE DYNAMISM

To use DYNMO with different example cases of dynamic models, we profile the dynamism mechanism for each example case. By large, the mechanism for measuring the load balance, redistributing the load, and re-packing (when possible) does not vary from case to case. DYNMO operates as a black-box approach where the load balancing happens at regular fixed intervals, without any knowledge of whether the model has changed or not. As will be shown in the results section, the very low overhead allows DYNMO to be invoked even at the granularity of each iteration. More details on the dynamism in the example use cases, and the dynamic reconfiguration of the pipeline available in Appendix F.

## 3.3 LOAD BALANCING

DYNMO implements two load balancing algorithms, and can be extended to support other algorithms. The first is centralized parameter-based partitioning that balances partitions based on the number of parameters. The load balancing algorithm is built on top of DeepSpeed's load balancing utility functions for partitioning in model parallelism Smith (2023). The second algorithm is an iterative decentralized diffusion-based algorithm that aims to minimize the variance between the workload of each rank by attempting to move layers from overloaded workers to underloaded ones in an iterative way. The workload cost can be described by either the layer execution times, or the parameter counts as in the centralized partitioning method.

The diffusion-based load balancing algorithm achieves ideal load balancing by iteratively minimizing workload imbalances using a Lyapunov-inspired approach. The potential function $\phi$, defined as the sum of workload gaps between workers, serves as a measure of imbalance in the system. Each iteration of the algorithm reduces $\phi$ by redistributing tasks from overloaded to underloaded workers, prioritizing layer transfers that yield the largest reductions in imbalance while satisfying memory constraints. The algorithm's probabilistic analysis guarantees that $\phi$ decreases towards a convergence threshold $\gamma$, with the rate of convergence bounded by $O(\mathcal{N}^2 \log(S\mathcal{N}/\gamma) \log \mathcal{N})$, where $\mathcal{N}$ is the number of workers and $S$ is the total pipeline size. This systematic reduction ensures that workload imbalances, quantified by the bubble ratio, are minimized, driving the system towards

an optimally balanced state. The theoretical guarantees of convergence and imbalance reduction underpin the algorithm's robustness in dynamic environments.

We demonstrate that the two load balancing schemes (used in Algorithm 1) meet the goals for optimal load balancing by using the following lemmas. Lemmas proofs presented in Appendix C.

**Lemma 1.** *A centralized load balancer $L_c$ over $\mathcal{N}$ workers satisfies maximum reduction in the imbalance $\mathcal{N}_i$ if and only if $\mathcal{N}_i$ reduces the bubble ratio to minimum.*

**Lemma 2.** *An iterative decentralized diffusion based load balancer $L_d$ over $\mathcal{N}$ workers satisfies maximum reduction in the imbalance $\mathcal{N}_i$ if and only if $\mathcal{N}_i$ reduces the bubble ratio to minimum. Also the load balancer is guaranteed to converge to the maximum reduction in imbalance in the following number of rounds*

$$O\left(\min\left\{\mathcal{N}^2 \log\left(\frac{S\mathcal{N}}{\gamma}\right) \log \mathcal{N}, \frac{S\mathcal{N} \log \mathcal{N}}{\gamma}\right\}\right)$$

where $\gamma \in \mathbb{R}_{>0}$ is the convergence factor and $\in \mathbb{R}_{>0}$ is the total number of stages in the pipeline.

### 3.4 RE-PACKING DYNAMIC MODELS TO FEWER WORKERS

Workload re-packing is the process of merging the total workload into a smaller number of workers (GPUs) with the purpose of using the available resources more efficiently, i.e. unused GPUs are released when the overall amount of work in training drops. This can be achieved with simple algorithms (in small scale) such as first-fit, best-fit, and round-robin as well as complex optimization heuristics. Workload re-packing aims to increase the worker utilization and reduce the overall number of workers employed to continue the training process. For long training schedules that are common in LLM training, workload packing can result in substantial cost savings. It may also provide improved performance due to reduction in the number of cross-worker communication calls, and smaller pipeline bubbles.

**Re-packing Definition** Re-packing occurs when the total workload $C^{(k)}$ decreases below a threshold $\tau$. At this point, the model layers (tasks) are redistributed across a smaller subset of workers to optimize resource utilization. Let $\mathcal{N}^{(k)} \subseteq \mathcal{N}$ denote the active set of workers after re-packing at time step $k$, where $|\mathcal{N}^{(k)}| \leq n$.

The load assignment function after re-packing is denoted by $A^{(k)} : \mathcal{T} \to \mathcal{N}^{(k)}$. The load of an active worker $N_i \in \mathcal{N}^{(k)}$ is defined as:

$$L_i^{(k)} = \sum_{t_j \in (A^{(k)})^{-1}(N_i)} c_j^{(k)}.$$

The objective after re-packing is to minimize the maximum load across the active workers:

$$\min_{A^{(k)}, \mathcal{N}^{(k)}} \max_{i \in \{1, \ldots, |\mathcal{N}^{(k)}|\}} L_i^{(k)}.$$

**Re-packing Condition** Re-packing is triggered if:

$$C^{(k)} < \tau \quad \text{and} \quad |\mathcal{N}^{(k)}| < n.$$

The aim of re-packing is to reduce the number of active workers $|\mathcal{N}^{(k)}|$, while ensuring that the maximum load remains balanced:

$$\max_{i \in \{1, \ldots, |\mathcal{N}^{(k)}|\}} L_i^{(k)} \approx \frac{C^{(k)}}{|\mathcal{N}^{(k)}|}.$$

We use a first-fit algorithm for workload consolidation. The goal of this algorithm is to reduce the number of active workers (subject to memory capacity constraints). When the combined memory usage of every pair of workers is less than the memory capacity of a single worker, we migrate the layers in order to free one of the GPUs. This repeats in an iterative fashion for every two pairs until no more GPUs can be eliminated. Appendix C elaborates on our algorithm for efficient re-packing.

When possible, repacking is an additional advantage to load balancing. We implement repacking in DYNMO at the load balancer level to enable the release of GPUs. To ensure the GPUs are released in a practical manner, we use the NCCL communicator splitting functionality to allow idle GPUs to be used by a concurrent communicator of the new job without the risk of deadlock. We elaborate on our method for releasing GPUs and list other alternative methods in Appendix C. DYNMO's task ends at releasing the GPUs; reclamation of the released GPUs and assigning them new jobs by the middleware or scheduler is outside the scope of this paper. That being said, it is worth noting that in single-node multi-GPU systems, Nvidia Multi-Instance GPU (MIG) Nvidia (2023) supports node partitioning for multi-tenancy. GPUs that have been released can be returned to MIG for allocation to other tenants. In multi-node environments, cloud schedulers have the ability to acquire released resources and reassign them to other jobs, often leveraging technologies like elastic Kubernetes Elastic (2023).

## 3.5 OVERHEAD OF DYNMO AND FREQUENCY OF DYNAMISM

The overhead of DYNMO is negligible. For all the results we show in this paper, for all model sizes and different dynamism example cases, the percentage of overhead is a few single digits at its highest; this includes the complete overhead of DYNMO: profiling data collection, rebalancing algorithm, and layers migration between GPUs. The evaluation section reports the load balancing overheads.

As a basic rule, we apply the rebalancing (via DYNMO) every time the model or the control flow changes. Since the overhead of DYNMO is very low, we could apply it as frequently as needed, based on the requirements of the applications. For example in gradual pruning the typical frequency of dynamism (i.e. model changes requiring load rebalancing) is in the order of 1,000s of iterations. On the other hand, for MoEs and MoDs the rebalancing in every iteration since the imbalance is unpredictable and dependent on point the routing decision is taken, i.e., in the forward pass at each FFN. For MoEs and MoDs we rebalance in the back propagation phase where we attach the movement of layers to the pipeline parallelism scheme (i.e. we migrate the layers as we propagate the gradients back down the pipeline from the last layer to the first layer).

## 3.6 LIMITATIONS

We focus on dynamism for LLMs and currently do not support other types of DNNs. Our method cannot automatically detect imbalance in a way that is transparent to the training process, i.e., we apply DYNMO at a fixed frequency regardless of whether the model has changed or not. We also allow the user to set the frequency. As future work, a background light weight tracker can be used to identify the dynamism pattern. Another alternative would be training our method to forecast the imbalance during training.

## 4 EVALUATION

This section contains empirical results and analysis of DYNMO's effectiveness. Elaborate details on the software, hardware, and training environment in Appendix A.

We conducted experiments using two dynamic load balancing algorithms, each with two different configurations. These algorithms were employed consistently the experiments for all six example cases of dynamic models. The first algorithm, referred to as *Partition: by Param*, is based on a DeepSpeed Rajbhandari et al. (2020) API. It uses a combination of binary search and linear probing to determine the optimal partitioning based on the parameter counts of the decoder layers. Another variation of this algorithm, called *Partition: by Time*, employs execution times of decoder layers as input. The second algorithm is a decentralized iterative diffusion-based load balancing approach,

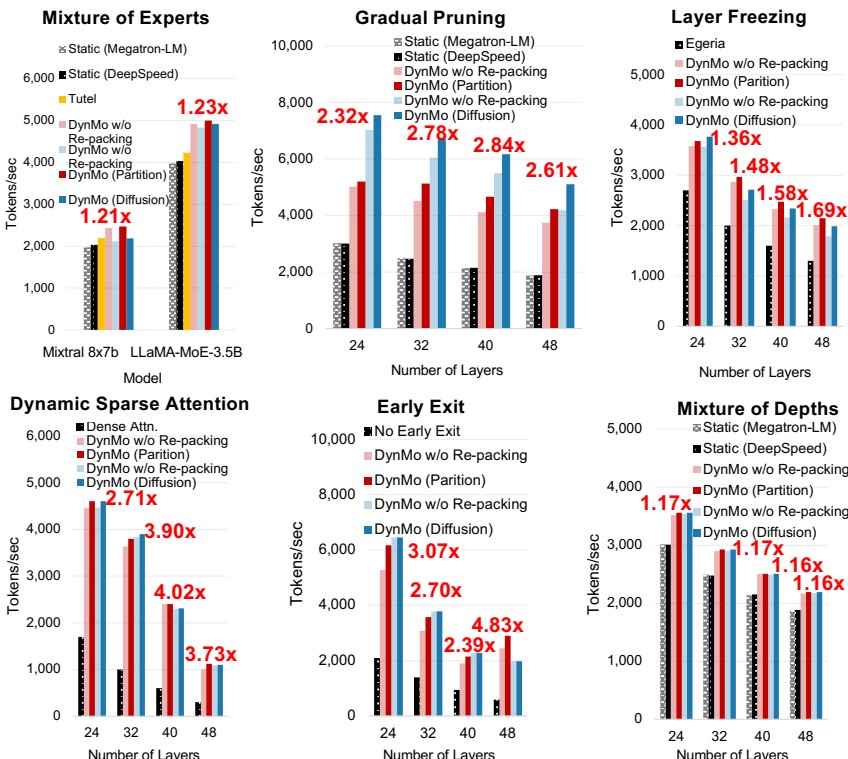

Figure 3: Throughput of end-to-end training for six different example cases. DYNMO rebalances at regular intervals w/o prior knowledge of dynamism. MoE, Sparse Attn., and MoDs: invoke DYNMO every iteration. Pruning, layer freezing, and early exit: invoke DYNMO every 100s to 1,000s iterations. Speedup we report is the highest among balancing by number of parameters or layer execution time, divided by the highest among static Megatron-LM and DeepSpeed (or SoTA baseline, when available). MoEs and MoDs: we use 128 GPUs (16 nodes each with 4x H100s) in a hybrid of 8-way data parallel + 16-way pipeline). For gradual pruning, layer freezing, dynamic sparse attention, and early exit we use a total of 720 H100 GPUs (90 nodes each with 4x H100s) in a hybrid of 30-way data parallel + 24-way pipeline.

which iteratively minimizes load variances among workers. Similar to DeepSpeed, this balancer has two variants: *Diffusion: by Param* and *Diffusion: by Time*.

## 4.1 END-TO-END TRAINING THROUGHPUT AND SPEEDUP

All our throughput and speedup results include the load balancing overhead, unless specified otherwise. We trained GPT models Radford et al. (2018) having different numbers of layers to determine the training throughput and speedup over static Megatron-LM and DeepSpeed (or SoTA baseline, when available). Figure 3 for MoEs, gradual pruning, and MoDs presents the highest throughput of two static and four dynamic load balancers. The first static balancer, Megatron-LM Shoeybi et al. (2019), evenly distributes layers across accelerators. The second static balancer, DeepSpeed Microsoft (2023), balances the number of parameters before training begins. In contrast, each of the two the dynamic load balancers (*Partition* and *Diffusion)* has two variants to redistribute the layers after each dynamism step: redistribute based on number of parameters or based on the layer execution time. Parameter-based balancers require profiling after the pruning step for memory usage information, while time-based balancers require profiling for memory usage and layer execution time information. We report the highest among both. Figure 3 for layer freezing, dynamic sparse attention, and early exit compare the two dynamic load balancers (*Partition* and *Diffusion)* over SoTA baseline that exists for those example cases. We observed that the use of layer execution time for dynamic load balancing, such as diffusion or partitioning, consistently outperforms parameter count-based implementations across all scales. In every scale, execution time-based dynamic balancers surpass the baseline static balancers. As seen in the figure, most of the speedup reported

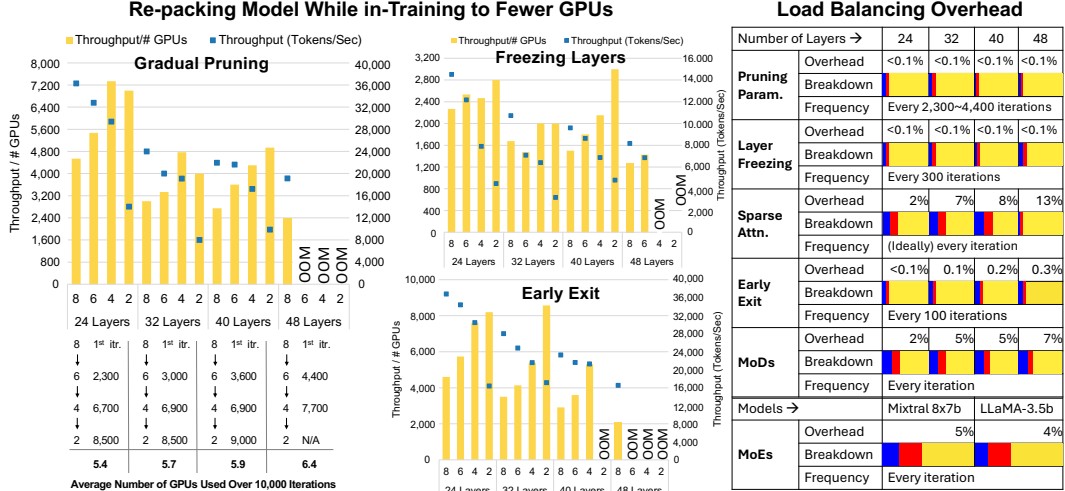

Figure 4: Re-packing the layers of GPT models into fewer GPUs as the model gets smaller due to dynamism. X-axis: Number of GPUs in pipeline parallelism using 90 nodes and up to 8 GPUs per node. Left Y-axis: throughput/number of GPUs shown with yellow bars. Right Y-axis: through-put (tokens/sec) shown with blue squares. Below: we show the average number of GPUs needed throughout the gradual pruning training at which we dynamically re-pack (total 10,000 iterations). Right: load balancing overhead for example cases. The overhead reported includes: profiling (in ▓), DYNMO load balancing algorithm (in ▓), and migration of layers between GPUs (in ▓).

is attributed to balancing the load by redistributing the, and not due to the reduced communication when we re-pack: the speedup gain from re-packing is between 4 11% of the entire speedup gain. In other words, even if we do not re-pack at, the speedup gain remains almost the same. Hence we treat re-packing as just a way to be efficient by using less GPUs, and not as a way to speed up imbalanced dynamic training. Additional experiments demonstrating the effect of network bandwidth, vertical scaling, and weak scaling are provided in Appendix E.

**Mixture of Experts** MoEs requires find-grained dynamism since the load vary from iteration to iteration. DYNMO shows more than 1.21x improvement on Mixtral 8x7b and LLaMA-MoE-3.5B in continual training. We do not change any of the hyperparameters from the original implementations. In addition to the Megatron-LM and DeepSpeed baselines, we also compare a highly MoE-tailored system: Tutel Hwang et al. (2022). DYNMO significantly outperforms Tutel: 1.18x on Mixtral 8x7b and 1.21x on LLaMA-MoE-3.5B.

The improvement margins on MoEs and MoDs are in fact among the top in all six use cases, relative to how much for the bubble ratio was eliminated. DYNMO reduces the bubble ratios of MoEs and MoDs from 25% to 8% and 18% to 4%, respectively. As a result, the end-to-end training of two production models improve 1.21x on Mixtral 8x7b and 1.23x on LLaMA-MoE-3.5B. Those improvements would translate to significant cost savings, considering the huge cost of training those models (and other similar models).

**Gradual Pruning** The pruning region starts from iteration 3000 and continues until iteration 7000 and the model is pruned every 1000 iterations until the 90% target sparsity is reached. This corresponds to sparsity levels of 52%, 79%, and 90% after each pruning step. All other hyperparameters are the same as Megatron-LM. Using layer execution time for diffusion or partitioning dynamic load balancing outperforms the parameter count-based implementations in each scale, for up to 3.18x.

**Layer Freezing** DYNMO ourperforms the SoTA layer freezing tool Egeria Wang et al. (2022). We can observe two main points. First, the speedups of different load balancing algorithms over static algorithms are largely similar. This is mainly because the different algorithms tend to arrive at similar load balancing solutions when entire layers are frozen. Second, DYNMO shows increased

speedup as the number of layers increases, particularly with diffusion, primarily because Egeria's overhead grows fast with the number of layers, while DYNMO overhead remains almost flat.

**Dynamic Sparse Attention** DYNMO achieves between 2.71x-4.02x speedup over the baseline dense attention (w/o sparsification). Dynamic sparse attention is the example case where DYNMO is most efficient at removing the pipeline bubbles. That is since using layer time execution, which fluctuates a lot in dynamic sparse attention, enables effective redistribution of the layers.

**Early Exit** DYNMO achieves more than 4x on average over the baseline w/o exit, i.e. when all tokens pass through the entire model. Similar to dynamic sparse attention, early exit benefits the most from DYNMO due to the big variance in load between earlier and later layers.

**Mixture of Depths** Like MoEs, MoDs layer loads vary from iteration to iteration. DYNMO shows 1.17x improvement on the baseline in continual training. We suspect MoDs will in the future be able to benefit more from custom load balancers that leverage the knowledge of how MoEs is used in hybrid with MoDs.

### 4.2 OVERHEAD OF LOAD BALANCING

Figure 4 (right) reports the load balancing overheads. This includes both the load balancing decision and the actual transfer of the parameters and other data of the layers to be sent or received, e.g., row offsets and column indices in CSR format for gradual pruning, and gradients in the case of MoEs and MoDs. The overhead is generally negligible, hence giving the opportunity for the use of DYNMO in other forms of dynamic models, and not just the six example cases in this paper.

### 4.3 RE-PACKING MODELS TO FEWER GPUS

In the re-packing experiments, the training starts with 8 GPUs per node in pipeline parallelism. After a dynamism step, DYNMO attempts to re-pack the total workload into fewer GPUs while satisfying the memory capacity constraints. Figure 4 reports the throughput/number of GPUs for each model size where the model is packed into 6, 4, and 2 GPUs. The 8 GPU setting for each model size serves as a baseline where there is no re-packing. This measurement also corresponds to the performance per dollar metric as the cost is directly proportional to the number of GPUs used in training.

We observe that in all model scales, re-packing can allow the training to be continued with fewer GPUs which may result in significant cost savings. For example, in gradual pruning we reduce the GPU count from 8 to an average of 5.8 GPUs while sustaining the training throughout.

## 5 CONCLUSION

DYNMO is a load-balancing system for dynamic models where the loads of the workers change during training. DYNMO provides better load balance than state-of-the-art static load balancing approaches, which results in better efficiency and faster end-to-end training time. Empirical results for LLMs with six example cases of dynamic models show that DYNMO significantly improves the training throughput over the counterparts. We foresee that dynamic models will be more prominent in the future and dynamic load distribution will be of utmost importance.

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

## A    Software, Hardware, and Training Environments

Experiments were mainly conducted on a supercomputer at which each of the compute nodes contains 2x AMD EPYC 9654 96-Core/2.4GHz processors, and 4x NVIDIA H100 SXM5 (80GB) GPUs. The GPUs in the same node communicate with CPUs using PCIe Gen5 x16 per GPU, and NVSwitch amongst the GPUs (NVLink34 x6). The compute nodes are connected by InfiniBand NDR200 200Gbps x 4. We used CUDA 12.1, OpenMPI 4.0.7, and PyTorch 2.3.1 with NCCL 2.17.1 distributed backend.

We train the models on the Wikipedia dataset Foundation (2023). for Mixtral 7bx8 and LLaMA-MoE-3.5B we do continual training. All models used for training have a sequence length of 2048, a hidden size of 1024, 32 attention heads, and the models are trained with a micro-batch size of 2 and batch size of 64 for 10,000 iterations, unless specified otherwise. For the multi-node experiments, as we increase the number of workers (GPUs), we also increase the batch size to fix the number of micro batches to four times the number of GPUs in the pipeline, as suggested in Huang et al. (2019) to achieve good pipeline utilization.

## B    Bubble Ratio in the Static Model

In this Section we describe the theoretical bubble ratio that appears in the static model. The *bubble ratio* refers to the ratio of the idle time of devices when different workers (GPUs) stall while waiting for work to be available. Additional bubbles appear in the pipeline when the model become dynamic (due to the load imbalance). The bubble ratio for the almost zero bubble pipeline scheme Qi et al. (2024) we use in the paper is:

$$\frac{\left(3\left(\frac{P^2}{2} - P\right) + 6P - 2\left(\frac{P^2}{2} - P\right)/P - 8\right)T_C + 2\left(\frac{P^2}{2} - P\right)T_B}{(3P^2 - 2P)T_C + (2P^2 - 2P)T_B + P^2 T_F} \tag{1}$$

where $S$ is the number of pipeline stages, $B$ is the number of micro-batches (chunks) in a single iteration, $P$ is the number of workers used in the pipeline, $T_F$ is the time cost for a complete forward pass (all forward stages added together) divided by P, $T_B$ is the time cost for a complete backward pass (all backward stages added together) divided by P, and $T_C$ is the communication time for moving a from a worker to its neighbor for the un-overlappable portion of communication.

The bubble ratio is derived from the un-overlappable portions of communication $T_C$ and forward pass $T_B$ (numerator of Equation 1) from the entire end-to-end span of the pipeline (denominator of Equation 1), where $\left(\frac{P^2}{2} - P\right)$ is the gaps/stalls in the pipeline due to lack of components to overlap after the forward and backward passes of the two duplicate models have been overlapped.

Figure 5 illustrates how the bubbles attributed to the dynamic sparsification adds up to, and is different from, the inherent bubbles that are observed in the pipeline scheme.

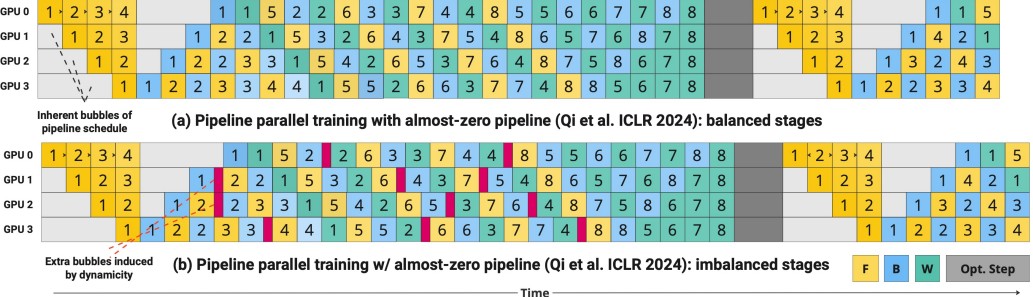

Figure 5: Illustration of bubble types in the almost zero bubble pipeline scheme Qi et al. (2024) with 8 microbatches. Each row represent a GPU's pipeline stages over time. Inherent bubbles in the pipeline are shown in gray and bubbles introduced by dynamicity (e.g. sparsity) are shown in red.

## C    PRUNING, LOAD BALANCING, AND PACKING

### C.1    NEURAL NETWORKS PRUNING

There are three main considerations that need to be taken into account when applying network pruning: criterion, structure, and schedule of the pruning.

**Pruning Criterion**: Every pruning scheme needs to define a criterion to choose which parameters to prune. A non-exhaustive list of pruning criteria used in the literature includes: weight magnitude Li et al. (2016); Renda et al. (2020), gradient magnitude Cun et al. (1990); Mozer & Smolensky (1989), Bayesian statistics-based criteria Dai et al. (2018); Molchanov et al. (2017), and reinforcement learning based criteria Lin et al. (2017); He et al. (2018). These criteria can be applied either locally (i.e. considering each layer's weights separately) or globally (i.e. considering weights in all layers).

**Pruning Structure**: Parameters in a model can be removed in a structured or unstructured way. Structured sparsity Kruschke & Movellan (1991) enforces a pattern to be applied while choosing the parameters to be pruned. This can range from removing filters in a convolution layer to removing attention heads in a multi-headed attention layer. On the other hand, unstructured sparsity Han et al. (2015) is not under the constraint of a pattern (i.e parameters can be freely removed), hence, offers a finer granularity. Even though unstructured sparsity offers better flexibility, structured sparsity is more prevalent since it is difficult to implement efficient kernels for sparse data structures in unstructured sparsity and deep learning frameworks have limited support for sparse computations. However, it has been shown that the enforcement of a certain structure for the pruning of parameters can result in significant degradation in model quality compared to unstructured sparsity Kalchbrenner et al. (2018); Elsen et al. (2020).

**Pruning Schedule**: After choosing the criterion and the structure of the pruning, one must decide when to prune and how often to prune. The most popular schedule in the literature consists of pruning after training is over, and then fine-tune the model to recover the loss introduced by the pruning Han et al. (2015). Another effective approach is to remove a certain percentage of weights progressively during the training until the target sparsity is reached Zhu & Gupta (2017), which eliminates the fine-tuning process. There are also schedules that enforce a constant rate of sparsity throughout the training Mocanu et al. (2018).

For a more comprehensive analysis of various sparsification procedures which are applied in deep learning, we refer the reader to Hoefler et al. (2021).

### C.2    GRADUAL GLOBAL MAGNITUDE PRUNING

For our pruning design, we use the gradual pruning schedule proposed in Zhu & Gupta (2017) which is formulated as:

$$S_t = S_f + (S_i - S_f)(1 - \frac{t - t_0}{n\Delta t})^3, \quad t \in \{t_0, t_0 + \Delta t, ..., t + n\Delta t\} \tag{2}$$

where $S_i$, $S_f$, $n$, $t_0$, and $\Delta t$ are initial sparsity, final sparsity, number of pruning steps, initial pruning step, and pruning frequency, respectively. The aim of this schedule is to prune the model rapidly in the initial pruning steps since there are many irrelevant connections, then reduce the pruning rate as the number of parameters in the network gets smaller.

We employed an unstructured magnitude pruning technique as opposed to a structured one since unstructured magnitude pruning typically retains better accuracy under high sparsity rates Prasanna et al. (2020). Unstructured magnitude pruning is applied globally (taking all parameters in the model into account) instead of locally since it has been empirically shown that global pruning yields better accuracy under high compression ratios Blalock et al. (2020).

To our knowledge, there is no deep learning framework that supports global pruning on a distributed model at the time of this writing (support is only for undistributed models). Thus we implemented our own global pruning algorithm as shown in Algorithm 2. The global pruning method takes three arguments, namely the model, target sparsity, and the rank of the device. Note that each rank[2] has

---

[2]We use one MPI rank per GPU.

---

**Algorithm 2** Dynamism (Global Pruning as an Example)

---

    **Input:** model, sparsity, rank
    **Output:** model
  1: params ← concat_params(model)
  2: k ← num_params × (1 - sparsity)
  3: local_topk, local_topk_indices ← topk(abs(params), k)
  4: topk_values ← gather(local_topk)
  5: **if** rank == 0 **then**
  6:     global_topk_indices ← topk(abs(topk_values), k)
  7: **end if**
  8: indices_to_keep ← scatter(global_topk_indices)
  9: model = compress_model(model, indices_to_keep)
10: **return** model

---

only its own portion of the model. First, each rank finds its own local top-$k$ parameters in terms of magnitude (line 3). Then, rank 0 gathers the top-$k$ parameters of each rank (line 4). When rank 0 receives all top-$k$ parameters, it calculates the indices of global top-$k$ parameters to keep (line 6), and sends the indices that belong to each rank (line 8). Finally, after each rank receives its indices to keep, they prune (discard) parameters with all other indices in their local parameters (line 9).

### C.3 LOAD BALANCING

DYNMO implements two load balancing algorithms, and can be extended to support other algorithms. The first one is a centralized parameter-based partitioning method that balances partitions based on the number of parameters. We also implemented a version where the same algorithm is used for balancing partitions based on the layer execution times instead of the number of parameters. This algorithm with two variants is built on top of DeepSpeed's load balancing utility functions for partitioning in model parallelism Smith (2023). The second algorithm is an iterative decentralized diffusion-based algorithm that aims to minimize the variance between the workload of each rank by attempting to move layers from overloaded ranks to underloaded ranks in an iterative way. The workload can either be the layer execution times or the parameter counts as in the DeepSpeed-based algorithms. The number of iterations to decide on the final load distribution is a user-defined parameter.

Algorithm 3 shows the pseudo-code for the diffusion-based load balancing algorithm. At the start of the balancing process, each worker evaluates their workload. A worker is deemed overloaded if their total workload exceeds the average workload across all workers and underloaded otherwise. Overloaded workers aim to offload some tasks to underloaded workers to achieve a more balanced distribution.

The balancing process unfolds iteratively. In each iteration, overloaded workers identify a task that contributes the least to their workload (e.g., a computational layer with minimal execution time). This task is then considered for transfer to the underloaded worker with the lightest workload. For every potential transfer, the algorithm computes the new workload distribution and evaluates the variance. A transfer is accepted if it reduces the variance and satisfies the memory constraints of the receiving worker. Accepted transfers are tracked, ensuring that tasks are redistributed efficiently.

By iteratively reducing workload variance through these localized decisions, the algorithm ensures a progressively balanced distribution. After the balancing phase, the transfer information is distributed to all workers, who update their local task assignments accordingly. This iterative and decentralized approach allows for effective load balancing in dynamic, distributed systems.

We elaborat on the details of the algorithm in this paragraph. After rank 0 gathers the loads (i.e. layer execution times or the number of parameters for each layer) from all ranks, it discovers all layer transfers between ranks by calling a diffusion re-balance function. The number of iterations to minimize the variance is an argument that can be tuned according to the workload. For each iteration of balancing, the total load of each rank, variance, and average load are calculated (lines 3-5). Then, each rank is assigned a status: *overloaded* or *underloaded* (lines 6-7). After the status of each rank is assigned, each overloaded rank attempts to send its least loaded layer to the least loaded

---

**Algorithm 3** Diffusion-based Load Balancing Algorithm

---

    **Input:** loads, num_ranks, max_iters, times, mem_info
    **Output:** transfers (list)
  1: transfers ← []
  2: **for** iter ← 0 **to** max_iters **do**
  3:      total_loads ← [sum($t$) for $t$ in times]
  4:      avg_load ← average(total_loads)
  5:      var ← variance(total_loads)
  6:      status ← ["Overloaded" if $l$ > avg_load
  7:             else "Underloaded" for $l \in$ loads]
  8:      **for** src ← 0 **to** num_ranks **do**
  9:         **if** status[src] == "Overloaded" **then**
10:            dst ← get_least_loaded_rank(loads)
11:            lyr_idx ← get_least_loaded_layer(src, times)
12:            new_loads ← update_loads(src, dst, lyr_idx, loads)
13:            new_total_loads ← [sum($l$) for $l \in$ new_loads]
14:            new_var ← variance(new_total_loads)
15:            mem_req = sum(mem_info[dst]) +
16:                 mem_info[src][lyr_idx]
17:            **if** new_var < var && mem_req < MAX_MEM **then**
18:               var ← new_var
19:               loads ← new_loads
20:               update_mem_info(src, dst, lyr_idx, mem_info)
21:               transfers.append((src, dst, lyr_idx))
22:            **end if**
23:         **end if**
24:      **end for**
25: **end for**
26: **return** transfers

---

rank (lines 7-24). Every time an overloaded rank attempts to send a layer to an underloaded rank, new loads and variance are calculated (lines 12-14). If the new variance is smaller than the current variance and it satisfies the memory constraints of the destination rank, the transfer is accepted and added to the transfers list in the format of (source, destination, layer id) (lines 17-22). When rank 0 discovers all layer transfers from source ranks to destination ranks, it distributes the information to other ranks and the sparse format data structures, CSR, of the layers to be transferred are sent to their new destinations.

We now demonstrate that the two load balancing schemes (used in Algorithm 3) meet the goals for optimal load balancing by using the following lemmas. Detailed proofs on the lemmas are presented in the supplementary material.

C.3.1   PROOF OF LEMMA 1

**Lemma 1.** *A centralized load balancer $L_c$ over $\mathcal{N}$ workers satisfies maximum reduction in the imbalance $\mathcal{N}_i$ if and only if $\mathcal{N}_i$ reduces the bubble ratio to minimum.*

*Proof.* We will prove by contradiction. Suppose a centralized load balancer $L_c$ over $\mathcal{N}$ workers satisfies maximum reduction in the imbalance $\mathcal{N}_i$ when $\mathcal{N}_i$ has a bubble ratio higher then the minimum. By the definition of maximum reduction in load balance, $L_c$ must preserve minimum differential between the loads of workers $\mathcal{N}_i$ and $\mathcal{N}_j$, which $\mathcal{N}_i$ and $\mathcal{N}_j$ have the minimum load and maximum loads in $\mathcal{N}$, respectively. Consequently, increasing the bubble ratio of $\mathcal{N}_j$ changes the difference of loads between $\mathcal{N}_i$ and $\mathcal{N}_j$. This is in contradictory of $L_c$ achieving the maximum reduction on imbalance. □

C.3.2   PROOF OF LEMMA 2

**Lemma 2.** *An iterative decentralized diffusion based load balancer $L_d$ over $\mathcal{N}$ workers satisfies maximum reduction in the imbalance $\mathcal{N}_i$ if and only if $\mathcal{N}_i$ reduces the bubble ratio to minimum.*

*Also the load balancer is guaranteed to converge to the maximum reduction in imbalance in the following number of rounds*

$$O\left(\min\left\{\mathcal{N}^2\log\left(\frac{S\mathcal{N}}{\gamma}\right)\log\mathcal{N}, \frac{S\mathcal{N}\log\mathcal{N}}{\gamma}\right\}\right)$$

where $\gamma \in \mathbb{R}_{>0}$ is the convergence factor and $S \in \mathbb{R}_{>0}$ is the total number of stages in the pipeline.

*Proof.* We leverages core ideas from Lyapunov optimization. We first define a potential function, $\phi$, that measures at each round the total magnitude of workload gaps in the system:

$$\forall r \geq 0 : \phi(r) = \sum_{u,v \in V} |x_u(r) - x_v(r)|$$

Similar to a Lyapunov function, $\phi$ maps the system state (in this case, a vector of workloads for $\mathcal{N}$ workers) at any given round to a non-negative scalar value that describes the desirability of the current system state. As $\phi$ decreases toward $0$, the system state becomes more desirable; i.e. the workload is balanced across $\mathcal{N}$. As in a standard Lyapunov optimization, we show below that the modifications to a system state caused by executing a single round of our max neighbor algorithm will drift the value of $\phi$ toward zero in a non-decreasing manner. We establish a probabilistic lower bound for the amount of drift in a given round to obtain our time bounds.

For a given round $r \geq 0$ and node pair $u, v \in V$, we define $d_{u,v}(r) = |x_u(r) - x_v(r)|$ to describe the gap between $u$ and $v$'s workload at the end of that round. For each such $r$, we also define: $\{\{u, v\} | u \text{ and } v \text{ connected and averaged their workloads in round } r\}$, i.e., the set of node pairs that connect and average in $r$, and $D_r = \sum_{u,v \in A_r} d_{u,v}(r-1)$, i.e., the sum of gaps averaged in $r$. Finally, we define $t_{max}(r) = max_{u,v \in V}\{d_{u,v}(r)\}$ to describe the largest gap between any two nodes at the end of round $r$. From the above analysis that $\phi(r)$ decreases by at least $D_r$ in each round $r$, we proceed to prove the converge time complexity bound.

For a maximum number of rounds to converge to the minimum imbalance:

$$O\left(\min\left\{\mathcal{N}^2\log\left(\frac{S\mathcal{N}}{\gamma}\right)\log\mathcal{N}, \frac{SN\log\mathcal{N}}{\gamma}\right\}\right)$$

Note that these two bounds essentially coincide at $\tilde{O}(\mathcal{N}^2)$ with $\gamma = \Theta(S/n)$, where the notation $\tilde{O}$ hides logarithmic factors. In other words, if we want all nodes to have the same workload up to a constant factor, the max neighbor strategy uses $\tilde{O}(\mathcal{N}^2)$ rounds. We first note that if we arrive at a round r in which $\phi(r) \leq \gamma$, then the system ends this round $\gamma$-converged, i.e. the sum of the gaps is at most $\gamma$, and thus clearly any individual gap is at most $\gamma$. Since $\phi$ is monotonically non-increasing, it follows that every round $r' \geq r$ is also $\gamma$-converged. So we just need to show that with high probability, $\phi$ will decrease to $\gamma$ in the time bound stated by the theorem statement.

For each $r \geq 1$, we call r "good" if and only if $\phi(r-1) - \phi(r) \geq s_{max}(r-1)/(60\, ln(2n))$. We next calculate how many good rounds guarantee that $\phi$ falls below $\gamma$. To do so, we first note that, non-good rounds cannot increase $\phi$, so we are safe to focus only on reductions generated by good rounds in calculating our bound.

By the definition of $\phi$, for each $r \geq 1$ we know that $\phi(r) < s_{max}(r)n^2$. It follows that if $r$ is a good round, then it decreases $\phi(r-1)$ by a multiplicative factor less than $(1 - \frac{1}{60n^2ln(2n)})$. Finally, we also observe that $s_{max}(0) \leq S$ and therefore $\phi(0) < Sn^2$. Leveraging these observations, to find the number of good rounds needed to decrease $\phi$ below $\gamma$, we just need to find the minimum $s$ time steps such that

$$Sn^2\left(1 - \frac{1}{60n^2ln(2n)}\right) \leq \gamma$$

A simple calculation implies that $s_{con} = 60n^2ln(2n)ln(Sn^2\gamma^{-1})$ is sufficient to satisfy this inequality. We have now established that after $s_{con}$ good rounds the system will be $\gamma$-converged for all future rounds. We are left to bound the number of rounds required to generate $s_{con}$ good rounds with high probability.

---

**Algorithm 4** Re-pack Layers into Fewer Workers

---

    **Input:** active_workers, mem_usage
    **Input:** target_num_workers, num_layers
    **Output:** transfers (list)
1: transfers ← []
2: **for** src in range(num_ranks) **do**
3:     **for** dst in range(src + 1, num_ranks) **do**
4:       **if** mem_usage[src] + mem_usage[dst] < MAX_MEM
5:     && sum(active_workers) > target_num_workers **then**
6:         active_workers[src] = 0
7:         **for** lyr_idx in range(num_layers[src]) **do**
8:           transfers.append((src, dst, lyr_idx))
9:         **end for**
10:        mem_usage[dst] += mem_usage[src]
11:        num_layers[dst] += num_layers[src]
12:       **end if**
13:     **end for**
14: **end for**
15: **return** transfers

---

For each round $r$, let $X_r$ be the random indicator variable that evaluates to 1 if round $r$ is good and otherwise evaluates to 0. We know a given round $r$ is good with probability at least $1/\mathcal{N}$, regardless of the history of the execution through the round $r - 1$. We cannot, however, directly leverage this observation to calculate (and concentrate) the expected sum of $X$ variables for a given execution length, as the distribution determining a given $X_r$ might depend in part on the outcome of previous experiments. To overcome this issue, we define for each round $r$, a trivial random indicator variable $\hat{X}_r$ that evaluates to 1 with independent probability $1/N$ and otherwise evaluates to 0. Note that for each $r$, $X_r$ stochastically dominates $\hat{X}_r$, and therefore for each $s > 0$, $Y_s = \sum_{r=1}^{s} X_r$ stochastically dominates $s > 0, \hat{Y}_s = \sum_{r=1}^{s} \hat{X}_r$. It follows for any $s > 0$, if $\hat{Y}_s \geq s_{con}$ with some probability $p$ then $Y_s \geq s_{con}$ with probability at least $p$.

A Chernoff bound applied to $\hat{Y}_s$, for $s = c.s_{con}$ (where $c \geq 1$ is a sufficiently large constant defined with respect to the constants in $s_{con}$ and the constants in the Chernoff form used), provides that $\hat{Y}_s \geq s_{con}$ with high probability, and therefore so is $Y_s$. To conclude the proof, we note that $c.s_{con} \in O\left(\mathcal{N}^2 log(\frac{S\mathcal{N}}{\gamma})log\mathcal{N}\right)$, as required by the theorem $\gamma$ statement. □

### C.4 RE-PACKING DYNAMIC MODELS TO FEWER WORKERS

Workload re-packing is the process of merging the total workload into a smaller number of worker (GPUs) with the purpose of using the available resources more efficiently, i.e. unused resources can be released. This can be achieved with simple algorithms (in small scale) such as first-fit, best-fit, and round-robin as well as complex optimization problems (for large scale) such as ant colony optimization Dorigo et al. (2006) or genetic algorithms Dasgupta et al. (2013). Workload packing aims to increase GPU utilization and reduce the overall number of GPUs employed to continue the training process. For long training schedules that are common in LLM training, workload packing can result in substantial cost savings. It may also provide improved performance due to reduction in the number of cross-GPU communication calls, and smaller pipeline bubbles.

Algorithm 4 shows a first-fit algorithm that we used for workload consolidation. We iterate over all the available GPUs (lines 2-3) and check if the combined memory usage of the two GPUs is less than the maximum memory capacity of a single GPU, and the number of active GPUs is greater than the target number of GPUs *target_num_gpus* for packing (lines 4-5). If that is the case, we transfer all layers of the source GPU to the destination GPU (lines 7-8). Then, it updates the memory usage and the number of layers on the destination GPU accordingly. This process continues until all the available GPUs have been checked and processed. The goal of this algorithm is to reduce the number of active GPUs to the *target_num_gpus*, while also ensuring that the total memory usage remains within device limits.

## C.5 Releasing Unused GPUs after Re-packing

The NCCL communicator cannot be resized, and there is no support for having more than one NCCL rank per GPU in the same communicator. Solutions around that include:

- The solution implemented and used in DYNMO: While the NCCL communicator can not be resized, NCCL supports splitting the communicator (using the `ncclCommSplit()` function) into multiple sub-partitions, or even creating a single communicator with fewer ranks Nvidia. The GPUs to which the model is re-packed can be split to their own sub-communicator, while the sub-communicator of the idle GPUs can be assigned to a different concurrent communicator of the new job to which they will be assigned (concurrent communicator are allowed in NCCL Nvidia). Since the released GPUs would never be used in the sub-communicator of the repacked GPUs, there is no risk or need to mitigate deadlocks of multiple concurrent communicators.

- Since large-scale full training runs for a long time (days and weeks) and MTBF (Mean Time Before Failure) in large scale training is typically in the order of hours, restarting from checkpoints becomes inevitable. On restarting the training from a checkpoint, the size of the communicator could be set to the new number of fewer GPUs.

- Different NCCL communicators can be used in a hierarchical fashion (and it is often the case). To allow for repacking, one could use a hierarchy of communicators with the intention to repack to one branch of hierarchy, and then assign a new job to the GPUs in the idle branch of the hierarchy.

Finally, we would like to emphasize that the primary focus of the paper is load balancing to make the end-to-end training process faster. Re-packing (when possible) is an additional advantage to load balancing. We implement repacking in DYNMO at the load balancer level to enable the release of GPUs. We would like to point out that DYNMO's task ends at releasing the GPUs: reclamation of the released GPUs and assigning them new jobs by the middleware or scheduler is outside the scope of this paper.

## D Implementation

The DYNMO load balancing system was developed on top of NVIDIA Megatron Core 0.5.0[3]. Each component of DYNMO, namely hooking to dynamism point in model training, load balancing, and re-packing is implemented in a separate package for ease of use and extension.

One particular challenging case in the six example cases is gradual pruning. The reminder of other example cases did not require hard codes changes to be able to use DYNMO. We elaborate more on gradual pruning. Unstructured pruning requires a sparse storage format to compactly store, train, and transfer the pruned model. One of the most commonly used sparse formats is the compressed sparse row (CSR) format. Using a sparse matrix format requires dense matrix multiplication (DMM) operations to be converted to sparse counterparts (SpMM). Since PyTorch does not support computing the derivative of SpMM operations for backpropagation on a CSR tensor, we evaluated CSR-based SpMM implementations available for use on GPUs, namely cuSPARSE by Nvidia and Sputnik Gale et al. (2020). Figure 6 shows the performance of cuSPARSE and Sputnik against the dense counterpart (cuBLAS). The SpMM kernel of Sputnik outperforms cuSPARSE in all sparsity levels. This is mainly because Sputnik kernels were implemented by specifically considering the deep learning workloads, unlike cuSPARSE kernels that mainly target the HPC workloads, which often have more than 99% sparsity. It is also worth noticing that Sputnik starts to outperform cuBLAS after 75% sparsity. Thus, for sparse operations, we implemented PyTorch bindings for the CUDA kernels of Sputnik [4].

The gather and scatter operations in global pruning were implemented by employing NCCL Peer-to-Peer (P2P) send-receive operations instead of collective communication operations since the sizes of the objects to be sent (local_topk) and received (indices_to_keep) from each rank are different and other ranks do not have this size information to participate in the collective call.

---

[3]https://github.com/NVIDIA/Megatron-LM/releases/tag/core_v0.5.0

[4]The Sputnik bindings are made available at the following link: https://anonymous.4open.science/r/Torch-Sputnik-E926/README.md.

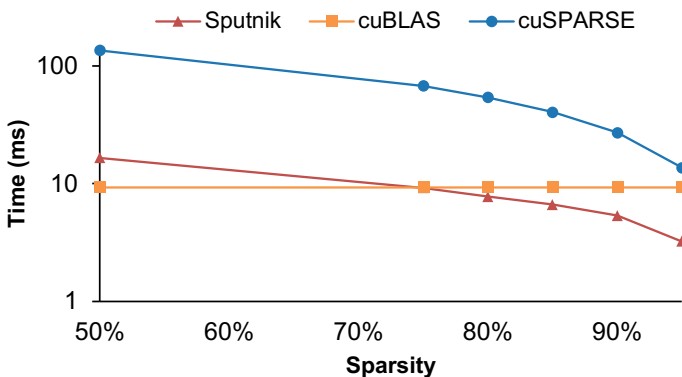

Figure 6: Sparse (Sputnik Gale et al. (2020) and cuSPARSE) vs Dense (cuBLAS) matrix multiplication performance comparison for M=N=K=4096 on Nvidia H100. Starting at 75% sparsity level, sparse kernels using Sputnik gives performance advantages over dense kernels.

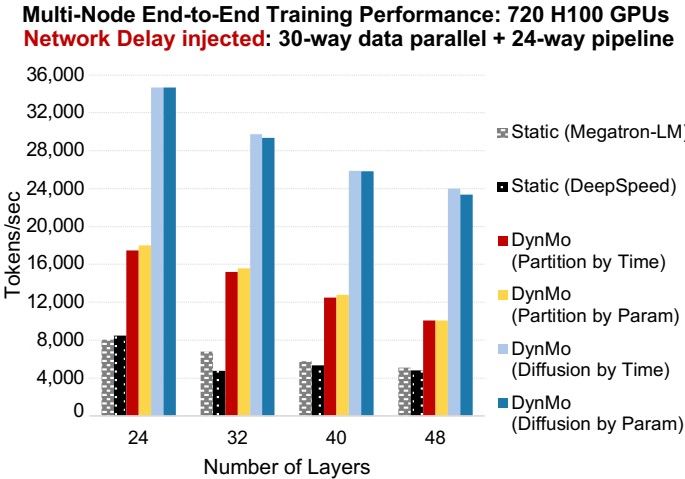

Figure 7: Multi-node end-to-end early exit training throughput of GPT models for a network with unstable bandwidth. We inject network delays up to 4x modeled by a normal distribution (as suggested by Sukhov et al. Sukhov & Kuznetsova (2009)). The intended delay is achieved by inflating the MPI message sizes based on the delay model.

The necessary information for load balancers such as layer execution times and memory usage comes from the profiling iteration after each pruning iteration. The execution time profiling is implemented by extending the built-in timers of Megatron-LM. The memory consumption of each pipeline stage is gathered with PyTorch's memory statistics for CUDA.

# E  ADDITIONAL RESULTS AND ABLATION

## E.1  EFFECT OF CHANGE IN NETWORK BANDWIDTH ON LOAD BALANCING

In fact DYNMO's load balancing algorithm is desgined to, indirectly, handle variability/instability in the network. That is since we define our diffusion load balancing algorithm to consider the gaps between workers to include the total time until work arrives to node B from neighboring node A, i.e. we include the amount of time that A spent on work plus the time it takes to transfer the activations of the layers over the network. In intuitive terms, if worker B is stalling due to delay from neighbor A (in part due to a slow network connection between A and B), the load balancer would push more work to worker B until the amount of work in A plus the time it takes to transfer the activations over the network is roughly equal to the amount of work on B.

Table 1: Vertical scaling experiments show the throughputs (samples/sec) of baseline Megatron-LM, and time-based algorithms, namely *Diffusion by Time* and *Partition by Time* for dynamic sparse attention. The speed is calculated for the best-performing balancer in each case. The benefits of dynamic load balancing increase as the number of GPUs in the pipeline increases. The number of GPUs listed are the GPUs in pipeline parallelism (in a 90-way data parallelism)

| # Layers | # GPUs | Megatron LM | Diff by Time | Part by Time | Speed Up |
|---|---|---|---|---|---|
| 24 | 2 | 10.67 | 12.38 | **12.88** | 1.20x |
| | 4 | 20.23 | **24.86** | 24.82 | 1.22x |
| | 8 | 37.253 | **46.939** | 45.071 | 1.26x |
| 32 | 2 | 8.28 | **10.35** | 10.12 | 1.25x |
| | 4 | 15.69 | 19.06 | **19.76** | 1.26x |
| | 8 | 30.809 | **39.899** | 37.933 | 1.29x |
| 40 | 2 | 7.14 | 8.84 | **9.13** | 1.28x |
| | 4 | 12.84 | 16.11 | **16.82** | 1.31x |
| | 8 | 26.425 | **35.262** | 33.296 | 1.33x |
| 48 | 2 | OOM | OOM | OOM | OOM |
| | 4 | 10.89 | **14.7** | 14.54 | 1.35x |
| | 8 | 23.126 | **31.724** | 29.711 | 1.37x |
| **56** | 2 | OOM | OOM | OOM | OOM |
| | 4 | OOM | OOM | OOM | OOM |
| | 8 | 19.126 | **26.724** | 25.711 | 1.39x |

Figure 7 shows results in a multi-node setting where we inject up to 4x delay in exchange of layers between neighbor nodes to demonstrate the robustness of DYNMO load balancing w.r.t. fluctuations in the network bandwidth. In fact the improvement of speedup of DYNMO over the baseline static model increases since the static model suffers from higher stalling when the network bandwidth fluctuates due to contention for instance.

## E.2 VERTICAL SCALING

In single-node multi-GPU vertical scaling experiments, the number of layers in the model and the number of GPUs used in the pipeline are changed. In Table 1, we report throughputs of the static baseline balancer (Megatron-LM) and the best-performing dynamic load balancers from end-to-end training experiments (*Diffusion by Time* and *Partition by Time*). The dynamic load balancers speed up the training in various degrees up to 1.39x for different numbers of GPUs.

One important observation is that as the number of GPUs used in the pipeline increases, the speedup gained by the usage of a dynamic balancer builds up. This suggests that the importance of load balancing increases as the pipeline gets deeper because the additional bubbles that are introduced by the dynamic nature of the model affect the efficiency of the pipeline more. This is important when considering the fact that the model size of large language models doubles approximately every 3.9 months Zhang et al. (2022) which leads to deeper pipelines.

## E.3 WEAK SCALING

For multi-nodes with multi-GPUs weak scaling experiments, we trained the GPT models having different numbers of layers and batch sizes (with work volume proportional to the number of GPUs) on up to 240 nodes, each of which contains 4x H100 GPUs. The pruning region starts from iteration 30 and continues until iteration 70 and the model is pruned every 10 iterations until the 90% target sparsity is reached. The pruning and load balancing over-heads are excluded from the measurements since the number of iterations to do this scaling experiment is not sufficient enough to amortize the overheads; in actual training (1000s to 10,000s iterations) the pruning and load balancing overheads would be negligible (elaborate overhead analysis in Appendix E.1). Figure 8 shows that the pipeline that is dynamically balanced with Partition by Time algorithm of DYNMO reaches higher throughputs in all scales and it provides speedups over baseline Megatron-LM up to 2.12x.

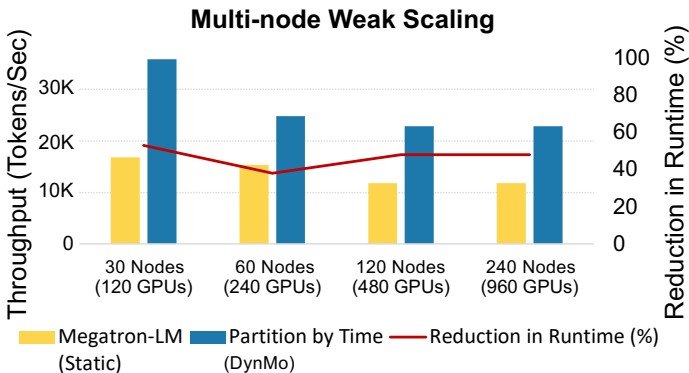

Figure 8: Gradual pruning weak scaling throughput (tokens/sec) comparison of baseline static load balancing with Megatron-LM and dynamic load balancing with *Partition by Time* algorithm of DYNMO.

### E.4 DYNAMIC MINIBATCH/MICROBATCH SIZE

In cases where the total load of the pipeline decreases such as gradual sparsification, dynamic sparse attention, early exit, and freeze training, carefully changing the minibatch and microbatch size according to the needs of the new pipeline after load balancing may increase the efficiency of the training. For instance, GPipe Huang et al. (2019) suggests the number of micro batches to be greater than four times the number of GPUs in the pipeline for optimal overlapping. Since the packing decreases the number of GPUs in the pipeline, adjusting the number of micro batches in the pipeline after packing could be beneficial. In addition, minibatch size can be increased after the pruning operations since the memory requirement for execution is less after the pruning. DYNMO currently does not support this feature, which if supported would further improve the speedup gains.

## F DYNAMISM

### F.1 DYNAMIC RECONFIGURATION OF THE PIPELINE

DeepSpeed's PipelineModule Microsoft currently offers three partitioning strategies for distributing model layers, which can be set using the `partition_method()` keyword argument passed to the `PipelineModule`. This allows us to move the layers between GPUs, when needed. When a layer is migrated from GPU A to GPU B, the memory allocated for the layer (weights, activation, gradients, optimizers) is released on GPU A and allocated on GPU B. As the training resumes (after the layers are migrated), the underlying pipeline scheme starts to pass the new mini-batches along the new distribution of layers in the pipeline.

### F.2 MIXTURE OF EXPERTS

We build on Mixtral 8x7b weights from Hugging Face Mistral (2024) and LLaMA-MoE-3.5B Team (2024) in continual training by monitoring the imbalance between layers w.r.t. number of assigned tokens. We used both the auxiliary load balance scheme adopted in Mixtral 8x7b Jiang et al. (2024) and the S-BASE load balancer Lewis et al. (2021a). Since the imbalance is dependent on the point the routing decision is taken, i.e., in the forward pass at each FFN, we rebalance in the back propagation phase where we attach the movement of layers to the pipeline parallelism scheme.

### F.3 GRADUAL GLOBAL MAGNITUDE PRUNING

For our pruning design, we use the gradual pruning schedule proposed in Zhu & Gupta (2017) which is formulated as:

$$S_t = S_f + (S_i - S_f)(1 - \frac{t - t_0}{n\Delta t})^3, \quad t \in \{t_0, t_0 + \Delta t, ..., t + n\Delta t\} \tag{3}$$

To our knowledge, there is no deep learning framework that supports global pruning on a distributed model at the time of this writing (support is only for undistributed models). Appendix C elaborates on the unstructured magnitude pruning scheme we implemented in PyTorch.

### F.4    LAYER FREEZING

DYNMO sits on top of layer freezing solutions. More specifically, we build on Wang et al. Wang et al. (2022) Egeria solution by monitoring the rate by which the training loss changes, freezing layers when they reach the convergence criterion, and drop frozen layers from in both the back propagation phase and gradient exchange when data parallelism is used. It is important to note that Egeria periodically updates the reference model (on the CPU) to drive the layer freezing, yet does not actively try to remedy the load imbalance caused by layer freezing. The effect of load imbalance is particularly pronounced since earlier layers tend to be more frozen than later layers, i.e. the layer freezing is not uniformly occurring across the model. In comparison, DYNMO load balances dynamically, and in an orthogonal fashion, the spread of layers on GPUs every time the reference model that drives the freezing is updated.

### F.5    DYANMIC SPARSE ATTENTION

We build on Pagliardini et al. dynamic sparse attention Pagliardini et al. (2023) in continual training by implementing a binding from PyTorch to the CUDA kernel provided by authors of the paper. The hash-based attention makes the causal attention mask irregular, i.e., we get blocked sparsity that is then leveraged by the Flash Attention. The irregular causal structures caused by the hashing lead to different amount of blocks/tiles in different layers.

### F.6    EARLY EXIT

We adapt the early exit methods CALM Schuster et al. (2022) and ADPC Liu et al. (2022b) to observe the imbalance by peaking into the confidence measure prediction of CALM and ADPC. Since early exit happens at the later layers, we start our observation from the first layer at which tokens start to exit, and we assume all layers before than to have the same load. Early exit in particular benefits greatly from re-packing, and that since the change in control flow of the model happens in the later layers.

### F.7    MIXTURE OF DEPTHS

We build on the MoDs work by Raposo et al. Raposo et al. (2024) by including in our GPT models we use in testing the small auxiliary MLP predictor that predicts whether that token will be among the top-k or not in the sequence. Similar to the case of MoEs, since the imbalance is dependent on the point the router takes the decision, i.e. in the forward pass, we rebalance in the back propagation phase where we attach the movement of layers to the pipeline parallelism scheme. Since the routing happens around the entire block, i.e., the routing applies to both to both forward MLPs and multi-head attention, we treat the skipped layers to be *shadow* layers when redistributing the layers on workers.

## G    RELATED WORK

### G.1    LOAD BALANCING MODEL-PARALLEL DEEP NEURAL NETWORKS

#### G.1.1    LAYER-WISE LOAD BALANCING

Layer-wise balancing techniques work on layer granularity instead of operators. DeepSpeed Microsoft (2023) offers three partitioning methods to balance the workload of stages: parameters, uniform, and regex. While the parameters method is trying to balance the number of parameters in each stage, the uniform aims to distribute the layers evenly. Regex only distributes the layers that match the given regex (e.g. transformer layers). Similar to the parameters method of DeepSpeed, He et al. He et al. (2021) balance the stages based on the number of parameters in each stage. Narayanan et al. Narayanan et al. (2021) assign each stage the same number of transformer layers to balance the

load. None of the aforementioned studies use the actual execution time of the layers to decide on the distribution of layers. DYNMO supports DeepSpeed's partitioning scheme with both parameters and layer execution times to guide load balancing, as well as a diffusion-based load balancing algorithm out of the box.

### G.1.2 LOAD BALANCING VIA GRAPH PARTITIONING

Graph partitioning-based load balancing schemes find atomic operations in the model and consider them as nodes in a directed acyclic graph (DAG). Edges in the graph represent the dependencies between operations. Tanaka et al. Tanaka et al. (2021) partition the DAG in three phases at which they first find atomic operations, then group these operations into blocks according to their computation times, and finally, they combine blocks into final partitions by using a dynamic programming-based algorithm. Qararyah et al. Qararyah et al. (2021) create disjoint clusters from the nodes of the graph by finding critical paths and mapping these clusters to devices based on a mapping algorithm that takes both critical-communication minimization and temporal load balancing into account. Both studies perform profiling before the actual training and partition the graph once.

### G.1.3 LOAD BALANCING IN MIXTURE OF EXPERTS MODELS

The mixture of experts (MoE) Jacobs et al. (1991) models contain many sub-networks (experts) where a router allocates inputs to top-k experts. At scale, experts are distributed across devices. Lepikhin et al. Lepikhin et al. (2020) defines an expert's capacity to limit the maximum number of tokens that can be processed by an expert to achieve workload balance. Fedus et al. Fedus et al. (2022) route each token to only one expert and use the same expert capacity for restrictions. Lewis et al. Lewis et al. (2021a) employ an auction algorithm Bertsekas (1992) to solve the token-to-expert assignment problem. This line of work is different from ours in the sense that their aim is to balance workload in the feed-forward network while our work aims to balance all layers of the transformer model.

## G.2 PACKING

In dynamic neural network models, packing the total workload into fewer number accelerators can provide significant cost-saving benefits. PipeTransformer He et al. (2021) offers an elastic pipelining system for freeze training where some of the layers of the model are frozen during the training. PipeTransformer packs the remaining active layers into fewer GPUs and creates pipeline replicas if possible. When PipeTransformer receives a notification for layer freezing, it attempts to divide the number of GPUs by 2 subject to the memory capacity constraints. On the other hand, our work DYNMO can pack to an arbitrary number of GPUs specified by the user. Another difference between the packing mechanism of DYNMO and PipeTransformer is that PipeTransformer uses the parameter size as a proxy to estimate the memory usage while DYNMO uses the actual memory usage from the profiling step before load balancing. Finally, PipeTransformer is only capable of packing layers to fewer GPUs, and not load balancing. DYNMO, on top of being capable of re-packing when deemed beneficial, it can also redistribute the workload to achieve a better load balance.

## G.3 DYNAMIC PRUNING

Model pruning is a fast-paced research area. Since the optimization problem has many dimensions, there are many approaches to prune a model. We mainly focus on the schedule of the pruning rather than the decision of how to prune (e.g. magnitude pruning, variational dropout etc.) and what kind of structure (e.g. unstructured pruning, structured pruning) to be applied while pruning.

One of the commonly used sparsification technique is sparsification during training (i.e. gradual pruning) where the pruning starts before the model is trained until convergence. While some studies Wortsman et al. (2019); Lin et al. (2020) use a binary mask to specify whether a parameter is pruned, which enables them to apply better weight regrowth or selection, others Gale et al. (2020) delete the pruned parameters to reduce the memory usage and number of operations. There are also many works on how fast to prune. For instance, Zhu and Gupta Zhu & Gupta (2017) prune the model rapidly in the first pruning steps when there are many abundant parameters in the model, and then reduce the pruning ratio as the number of parameters in the model are getting less and less. Dai et

al. Dai et al. (2019) employ a three phase schedule (birth-brain, baby-brain, and adult-brain) similar to the human brain development. Mostafa et al. Mostafa & Wang (2019) uses magnitude pruning as criterion to prune the parameters and regrows parameters to comply with the training budget.

