# OpenReview forum: "Elastic and Balanced End-to-end Training of Dynamic LLMs with DynMo"
_ICLR.cc/2025/Conference — Submitted to ICLR 2025_

### Official Review · Reviewer_f6tK · 2024-10-28

**Soundness:** 2
**Presentation:** 1
**Contribution:** 2
**Rating:** 3
**Confidence:** 3

**Summary:**

This paper discusses how to reduce the load imbalance during dynamic model training. Dynamic training is emerging to reduce the computational and memory costs of LLMs. However, it leads to workload imbalance among workers, which negatively affects training efficiency. This paper proposes DynMo that adaptively maintains equal compute workloads among workers in pipeline parallelism as well as packs work into fewer workers. The evaluation with six different example cases of dynamic models demonstrates the effectiveness of DynMo.

**Strengths:**

+ This paper discovers load imbalance in six example cases of dynamic training and measures their bubble ratios with pipeline parallelism.
+ The extensive evaluation shows DynMo can greatly reduce the bubble ratios and improve the end-to-end training efficiency without harming model accuracy.
+ The paper uses a production-level cluster for the evaluations

**Weaknesses:**

- The paper writing needs to be improved. 1/ Load imbalance is a major motivation for this paper, but there is no formal definition in the paper; 2/ This paper identifies six example cases for load imbalance, but it is unclear why these dynamic model training leads to load imbalance and worsens communication bubbles. More fundamental analysis is needed to understand the problem.
- The two algorithms introduced in Section 3.3 are not convincing to achieve the claimed goal for optimal load balancing. In addition, the detailed description of this critical algorithm is not included in the main body, but in the appendix. At least some intuitions for why it can achieve load balancing should be discussed in Section 3.
- From a system perspective, re-packing dynamic models to fewer workers requires to reshard the model checkpoint, recompute the graph, and restart the training process, which will incur costly overhead up to several minutes for LLMs. However, this paper claims the overhead of DynMo is negligible, without discussing how it minimizes these incurred system overheads. I think an implementation section is needed for fidelity.

**Questions:**

N/A

---

> ### Author Response · Authors · 2024-11-21
>
> >The paper writing needs to be improved. 1/ Load imbalance is a major motivation for this paper, but there is no formal definition in the paper; 2/ This paper identifies six example cases for load imbalance, but it is unclear why these dynamic model training leads to load imbalance and worsens communication bubbles. More fundamental analysis is needed to understand the problem.
>
> We added a formal definition of load balancing and re-packing as a general problem for dynamic models, and adapted it further to formally define the imbalance arising from each of the example cases. We added the formal definitions to the paper: Section 2.2 (line 168) for load balancing and Section 3.4 (line 611) for re-packing. Note: we added the above to the paper proper to help the reviewers locate the added content. should the paper be accepted, the added formal definitions will be only summarized in the paper proper, and the details would be moved to the supplementary material.
>
> >The two algorithms introduced in Section 3.3 are not convincing to achieve the claimed goal for optimal load balancing.
>
> The paper includes a full theoretical proof that the algorithms are guaranteed to converge to the maximum reduction in imbalance in bounded rounds (proofs of Lemma 1 and Lemma 2 in Appendix 3.2). We further explain in detail how the algorithm lead to convergence in Appendix 3. Finally, we show empirical results that the algorithms lead to the required ideal balance by eliminating the stalling in the GPUs (the speedups in Figure 3 show that the idealess we observed in Figure 1 is eliminated).
>
> >In addition, the detailed description of this critical algorithm is not included in the main body, but in the appendix. At least some intuitions for why it can achieve load balancing should be discussed in Section 3.
>
> We acknowledge that the readability can be improved by changing the structure to move the more important parts to the paper proper. We do however like to point out that is a structuring issue that could be easily addressed by re organizing the paper, and is not a fundamental issue with the methodology or results.
>
> >From a system perspective, re-packing dynamic models to fewer workers requires to reshard the model checkpoint, recompute the graph, and restart the training process, which will incur costly overhead up to several minutes for LLMs. However, this paper claims the overhead of DynMo is negligible, without discussing how it minimizes these incurred system overheads.
>
> There are two things that are separate here: redistributing the model layers to load balance, and re-packing. For registering the model layers, the overhead (inclusive of profiling and migrating layers) is very low (single digit percentage) and hence is not an issue. For re-packing, we iterate that re-packing happens at a much lower frequency than rebalancing. While rebalancing can happen at the granularity of iterations, in practice it takes 1000s of iterations for the LLM to change enough to the point where re-packing would be beneficial (See Figure 4). Accordingly, the re-packing can be aligned to happen with the model restarts from a checkpoint since MTBF in practice is lower than the pace by which the model is re-packed. This makes re-packing much simpler and come for free since we piggyback on the restart process.
>
> >I think an implementation section is needed for fidelity.
>
> Appendix D covers the implementation details. In addition the code of DynMo is open sourced, and it includes a detailed documentation.

---

> > ### Author Response · Authors · 2024-11-25
> >
> > We would be glad to answer any questions about the paper.

---

### Official Review · Reviewer_TGsS · 2024-11-02

**Soundness:** 3
**Presentation:** 4
**Contribution:** 3
**Rating:** 5
**Confidence:** 5

**Summary:**

DYNMO is a load-balancing system for dynamic models where the loads of the workers change during training. DYNMO provides better load balance than state-of-the-art static load balancing approaches, which results in better efficiency and faster end-to-end training time. DYNMO implements two load balancing algorithms built on top of DeepSpeed’s load balancing utility functions. The first is centralized parameter-based partitioning and the second algorithm is an iterative decentralized diffusion-based algorithm. Empirical results for LLMs with 6 example cases of dynamic models show that DYNMO significantly improves the training throughput over the counterparts.

**Strengths:**

Thank you for submitting this paper to ICLR!

1. Comprehensive evaluation. The study presents an extensive evaluation conducted on a large-scale GPU cluster with hudreds of H100 GPUs. This rigorous assessment demonstrates the scalability and effectiveness of the proposed approach.

2. Provide source code repository. This resource not only facilitates the replication of the presented results but also empowers others to leverage the system for improved training times through dynamic techniques.

3. Interesting topic. This paper addresses a issue involving the integration of pipeline parallelism with dynamic pruning and freezing.

**Weaknesses:**

1. Limited Technical Innovation: The technical contribution of the paper is somewhat restricted, as it relies on the application of established techniques in a novel context rather than introducing entirely new methodologies.

2. Unclear algorithm descriptions: The second algorithm is an iterative decentralized diffusion-based algorithm. For the "diffusion", it is not clear how the diffusion is performed. The paper should provide a more intuitive example / explanation of the algorithm.

3. Omission of a Dynamic Baseline Configure: I think Megatron and Deepspeed are also support some naive dynamic baseline configure (manually). The paper should compare with them as a stronger baseline.

**Questions:**

As a training framework, it is a bit wired to support elastic (I mean reduce GPU amount). Most work discusses fault-tolerance for LLM-pretraining (e.g., Gemini [SOSP '23], ReCycle [SOSP '24]). In other words, it is important to demonstrate the wide requirement of dynamic training (e.g., layer freezing).

1. I highly recommend changing the story to cluster-level scheduler instead of a training framework. It makes sense to automatically reduce some resources when some layer is freeze. Other jobs can better utilize the resource.

2. The experiment is solid, but if I focus on MoE training case, I want to see some comparison with MoE-tailored system like Tutel https://github.com/microsoft/tutel

---

> ### Author Response · Authors · 2024-11-21
>
> >Unclear algorithm descriptions: The second algorithm is an iterative decentralized diffusion-based algorithm. For the "diffusion", it is not clear how the diffusion is performed. The paper should provide a more intuitive example / explanation of the algorithm.
>
> We added an intuitive explanation to Appendix C.3 (line 1275).
>
>
> >Omission of a Dynamic Baseline Configure: I think Megatron and Deepspeed are also support some naive dynamic baseline configure (manually). The paper should compare with them as a stronger baseline.
>
> Megatron and DeepSpeed do not offer any functionality for dynamically balancing the model, not even with manual control, hence there is no baseline in Megatron or DeepSpeed to compare with (even with manual control). What DeepSpeed does simply offer is an API to dictate how to distribute the layer, which is in fact what we use to implement the redistribution of work load by DynMo.
>
>
> >I highly recommend changing the story to cluster-level scheduler instead of a training framework. It makes sense to automatically reduce some resources when some layer is freeze. Other jobs can better utilize the resource.
>
> Most of the speedup reported in this work is attributed to balancing the load by redistributing the, and not due to the reduced communication when we re-pack: the speedup gain from re-packing is between 4~11% of the entire speedup gain. In other words, even if we do not re-pack at, the speedup gains remain almost the same. Hence we treat re-packing as an additional benefit of DynMo to be efficient by using less GPUs, and not as a way to speed up imbalanced dynamic training. We modified Figure 3 to include speedups when re-packing is used versus without.
>
> >The experiment is solid, but if I focus on MoE training case, I want to see some comparison with MoE-tailored system like Tutel https://github.com/microsoft/tutel
>
> We added results that compare with Tutel in Figure 3 (line 727). DynMo significantly outperforms Tutel: 1.18x on Mixtral 8x7b and 1.21x on LLaMA-MoE-3.5B.

---

> ### Author Response · Authors · 2024-11-25
>
> We would be glad to answer any questions about the paper.

---

> > ### Comment · Reviewer_TGsS · 2024-12-01
> >
> > Thank you for your response. My concerns have been addressed. I will keep my score at the current rating.

---

### Official Review · Reviewer_QEEN · 2024-11-04

**Soundness:** 4
**Presentation:** 3
**Contribution:** 3
**Rating:** 8
**Confidence:** 3

**Summary:**

This paper presents a novel dynamic load balancing solution (DynMo) designed to address the computational and memory challenges associated with Large Language Models (LLMs). The authors identify that dynamic training schemes, such as Mixture of Experts (MoEs), parameter pruning, layer freezing, dynamic sparse attention, early exit strategies, and Mixture of Depths (MoDs), can lead to workload imbalance among workers in distributed training environments, negatively impacting efficiency.

The primary contributions of this paper are as follows:
1. A dynamic load balancing framework that adaptively maintains equal compute workloads among different workers in pipeline parallelism, ensuring balanced pipeline stages during training.
2. This paper provides a proof that DynMo satisfies the maximum reduction in imbalance, effectively mitigating the negative effects of dynamic models on pipeline utilization.
3. DynMo dynamically packs work into fewer workers while sustaining training throughput, allowing for the release of idle workers back to the job manager, which can lead to cost savings and improved resource utilization.

**Strengths:**

1. The authors have identified and formulated the problem of workload imbalance in dynamic LLMs, which is a relatively unexplored area. The formulation of this problem and the proposed solution are original contributions to the field.
2. This paper provides mathematical proofs that DynMo satisfies maximum reduction in imbalance, which adds a strong theoretical foundation to the work.
3. The framework's compatibility with various dynamic compute and model reduction schemes suggests that it could be widely adopted across different domains and models, increasing its significance.

**Weaknesses:**

As the author said in section 3.6, this paper only focuses on the end-to-end training problem of dynamic LLMs, and the proposed DynMo framework and method are only applicable to dynamic LLMs. In addition, the method proposed in this paper cannot automatically detect imbalance and must be manually configured by the user before training begins.

**Questions:**

I think the contribution of this paper is valuable and universal, so I suggest that the author can explain how their idea can be applied in an automated framework in the future, or even integrate it into existing mainstream frameworks.

---

> ### Author Response · Authors · 2024-11-21
>
> We thank the reviewer for the encouraging feedback.
>
> >In addition, the method proposed in this paper cannot automatically detect imbalance and must be manually configured by the user before training begins.
>
> No manual configuration is needed from the user: the imbalance is done on regular intervals, i.e. we do not detect the imbalance, we instead relay on the fact that the load balancing overhead is very low, and just apply the load balancing at regular intervals. There could be an opportunity to further improve by detecting the imbalance, but that would improve the performance marginally.
>
> >I think the contribution of this paper is valuable and universal, so I suggest that the author can explain how their idea can be applied in an automated framework in the future, or even integrate it into existing mainstream frameworks.
>
> This is indeed a priority for us. DynMo is fully automated, and requires no input from the user, hence can be add easily to existing frameworks. In fact we are actively trying to port DyNMo to use the new Megatron-core layer migration capability, and hence that would allow users who already use Megatron-core to use DynMo by adding a single line to their PyTorch code that activates the load balancing.

---

> > ### Comment · Reviewer_QEEN · 2024-11-22
> >
> > Thank you to the authors for addressing the issues I raised. I would prefer to maintain my previous score.

---

### Official Review · Reviewer_ubJG · 2024-11-04

**Soundness:** 3
**Presentation:** 2
**Contribution:** 2
**Rating:** 5
**Confidence:** 2

**Summary:**

To address workload imbalance in LLM training, this paper proposed a method to redistribute the workload. It composes well with dynamoc training technics including MoE/MoD, parameter pruning, layer freezing, and early exit. As workload shrinks, DynMo can pack work into fewer works and release idle workers for better GPU utilization.

In terms of redistribution overhead, the author shows that it's <10% of the training QPS, including profiling, making decision, and actual data transfer

**Strengths:**

a) open sourced: the proposed method is open sourced and is built on top of megatron. The instructions are clear and easy to follow
b) covers critical details: it explained how to overcome the fact that each gpu can only have 1 NCCL communicator. These are practical challenges we need to resolve to truly reuse the idle worker

**Weaknesses:**

a) it remains unclear how resouce are isolated for safty purpose. For example, if NCCL communicator are shared, how could we make sure main job are safe as 3rd party job can do intentionally use it for other purpose.
b) the imbalance detection overhead is proportional to profiling frequency and is highly empirical. The claim of single digit overhead is up to a specific job setting

**Questions:**

When repacking jobs into fewer workers, what are things we are serializing? Are they model states and optimizer states? Do all the workers stop at the same time and start repacking? How do we deal with NCCL timeout if some rank are faster?

When consolidating profiler results, how to measure the imbalance among different workers? If the imbalance is caused by a straggler rank, how do we deal with it?

---

> ### Author Response · Authors · 2024-11-21
>
> >a) it remains unclear how resources are isolated for safety purpose. For example, if NCCL communicator are shared, how could we make sure main job are safe as 3rd party job can do intentionally use it for other purpose.
>
> First of all: excellent point!
>
> Re-packing happens at a much lower frequency than rebalancing. While rebalancing can happen at the granularity of iterations, in practice it takes 1000s of iterations for the LLM to change enough to the point where re-packing would be beneficial (See Figure 4). Accordingly, the re-packing can be aligned to happen with the model restarts from a checkpoint since MTBF in practice is lower than the pace by which the model is re-packed. This makes re-packing much simpler (for example, we anyway launch an entirely new NCCL communicator upon restarting from a checkpoint).
>
> In addition, if we wish to not-pack upon model restart, and sandbox the sub-communicator, there are several options (beyond the scope of this paper, yet we list since we agree with the reviewer that this is an interesting problem to think about):
> - To add an extra layer of sandboxing: NCCL ranks validation to validate ranks within each communicator to ensure they only belong to the designated job.
> - Implement access control in the NCCL runtime: If modifying the implementation is an option, one can enforce sandboxing at the runtime level by: a) adding an interception layer to wrap NCCL calls with validation checks to ensure processes cannot reference other communicators, or b) use custom communicator IDs that assign unique identifiers to each communicator and restrict access to these IDs based on process membership.
>
> >When repacking jobs into fewer workers, what are things we are serializing? Are they model states and optimizer states?
>
> Yes: model parameters, model states, and optimizer states.
>
> >Do all the workers stop at the same time and start repacking? How do we deal with NCCL timeout if some rank are faster?
>
> We schedule the re-packing to happen at the end of iterations since there is already a barrier at the end of iteration. That way we don't need to introduce any additional barriers.

---

> > ### Comment · Reviewer_ubJG · 2024-11-27
> >
> > thanks for pointing out the differences between re-packing and rebalancing. It makes sense that NCCL communicator are inited upon restarting. I would like to keep my rating as it is.

---

> ### Author Response · Authors · 2024-11-25
>
> We would be glad to answer any questions about the paper.

---

### Author Response · Authors · 2024-11-21

We thank reviewers for the comments. We cover the common concerns here, and answer individual questions directly to each reviewer.

>Novelty and impact of DynMo

To the authors knowledge, DynMo is the first solution to address dynamic models training. We cover several example cases, including example cases for production training solutions (MoEs, layer freezing, and MoDs). DynMo is fully automated, and requires no input from the user, hence can be used with other types of dynamic models beyond the example cases in the paper. We demonstrate end-to-end training speedup gains from DynMo on ~1,000 GPUs scales up linearly. The overhead for profiling and redistributing the workload are single digit percentage in comparison to several factors of speedup: 1.23x (MoEs), 3.18x (parameter pruning), 2.23x (layer freezing), 4.02x (sparse attention), 4.52x (early exit), and 1.17x (MoDs).

>The role of re-packing, and its effect on speedup gains.

Most of the speedup reported in this work is attributed to balancing the load by redistributing the, and not due to the reduced communication when we re-pack: the speedup gain from re-packing is between 4~11\% of the entire speedup gain. In other words, even if we do not re-pack at, the speedup gains remain almost the same. Hence we treat re-packing as an additional benefit of DynMo to be efficient by using less GPUs, and not as a way to speed up imbalanced dynamic training. We modified Figure 3 to include speedups when re-packing is used versus without.

Finally, we iterate that re-packing happens at a much lower frequency than rebalancing. While rebalancing can happen at the granularity of iterations, in practice it takes 1000s of iterations for the LLM to change enough to the point where re-packing would be beneficial (See Figure 4). Accordingly, the re-packing can be aligned to happen with the model restarts from a checkpoint since MTBF in practice is lower than the pace by which the model is re-packed. This makes re-packing much simpler (for example, we anyway launch an entirely new NCCL communicator upon restarting from a checkpoint).

>Formal definition of load imbalance of dynamic models

We added a formal definition of load balancing and re-packing as a general problem for dynamic models, and adapted it further to formally define the imbalance arising from each of the example cases. We added the formal definitions to the paper: Section 2.2 (line 168) for load balancing and Section 3.4 (line 611) for re-packing. Note: we added the above to the paper proper to help the reviewers locate the added content. should the paper be accepted, the added formal definitions will be only summarized in the paper proper, and the details would be moved to the supplementary material.

---

### Meta-Review · Area_Chair_qnaC · 2024-12-24

**Metareview:**

This paper discusses how to reduce the load imbalance during dynamic model training, which is observed to have workload imbalance among workers, and negatively affects training efficiency. This paper proposes DynMo that adaptively maintains equal compute workloads among workers in pipeline parallelism as well as packs work into fewer workers. The evaluation with six different example cases of dynamic models demonstrates the effectiveness of DynMo.

The strength of the paper:
* address the dynamic LLM training, which is a less explored area
* proposed ideas are sensible

The weakness of this paper:
* The writing of this paper could be greatly improved. The initial paper looks like it was not polished and submitted in a hurry. The author added a lot of content during rebuttal, resulting in a 16-page paper. The paper's story and arguments could also be curated in a better way as a reviewer pointed out.
* The chosen baselines were weak. As several reviewers pointed out, there do exist system (variants) built for addressing each of the workloads targetted by Dynmo (e.g., MoE, sparse attention); the authors should consider comparing to those baselines.
* In industry practice, dynamic training and pipeline parallelism, which were targeted by this paper, are not a common combination, which diminishes the practical values of this paper.


The weakness outweighs the strength, hence a rejection.

**Additional Comments On Reviewer Discussion:**

The authors added a lot of content and clarification during the rebuttal. After rebuttal, 3 of 4 reviewers remain negative about this paper.

---

### Decision · Program_Chairs · 2025-01-22

Reject